# Emergent mechanics of a networked multivalent protein condensate

Zhitao Liao [1,6], Bowen Jia [2,6], Dongshi Guan [3,4] ✉, Xudong Chen[2], Mingjie Zhang [2,5] ✉ & Penger Tong [1] ✉

Multivalent proteins can form membraneless condensates in cells by liquid-liquid phase separation, and significant efforts have been made to study their biochemical properties. Here, we demonstrate the emergent mechanics of a functional multivalent condensate reconstituted with six postsynaptic density proteins, using atomic-force-microscopy-based mesoscale rheology and quantitative fluorescence measurements. The measured relaxation modulus and protein mobility reveal that the majority (80%) of the proteins in the condensate are mobile and diffuse through a dynamically cross-linked network made of the remaining (20%) non-mobile scaffold proteins. This percolating structure gives rise to a two-mode mechanical relaxation with an initial exponential decay followed by a long-time power-law decay, which differs significantly from simple Maxwell fluids. The power-law rheology with an exponent $\alpha \simeq 0.5$ is a hallmark of weak bonds' binding/unbinding dynamics in the multivalent protein network. The concurrent molecular and mechanical profiling thus provides a reliable readout for characterizing the mechanical state of protein condensates and investigating their physiological functions and associations with diseases.

Liquid-liquid phase separation (LLPS) has recently been recognized as an important organizing mechanism for various membraneless intracellular compartments and structures in living cells[1–3], such as membrane clusters[4], postsynaptic densities[5], P granules[6,7], stress granules[8] and nucleoli[9]. At certain concentrations, the proteins and other biomolecules can spontaneously form a condensed phase via LLPS[1–3,10,11]. At the molecular level, a variety of condensate-forming conditions[12,13], component-varying strategies[5] and delicate mutations[8,14] have been used to understand the biochemical nature of condensate formation and evolution. Numerical simulations and theoretical approaches were employed to provide physical insights into the molecular interactions in the condensates[15–19]. These studies laid down the foundation for our understanding on how the individual protein ingredients interact and induce LLPS against entropic mixing.

At the coarse-grained level, attempts were also made to study the material properties of the protein condensates. For example, several fluid parameters, such as viscosity $\eta$, surface tension $\gamma$, modulus $E$, and diffusion coefficient $D$, were obtained separately for simple protein condensates using the techniques of fluorescence imaging, particle tracking and optical tweezers[13,20–25]. Among the fluid parameters, the diffusion coefficient $D$ measures the protein mobility inside the protein droplets and tells us about the molecular crowdedness of the condensate[6,13,26,27]. In recent microrheology experiments[20,22], optical tweezers were used to measure the mechanical behavior of a submicron- or micron-sized particle seeded in a protein droplet, from which the frequency (or $\omega$) dependent complex shear modulus $G^{*}(\omega) = G'(\omega) + iG''(\omega)$ of the condensate was obtained.

[1]Department of Physics, Hong Kong University of Science and Technology, Clear Water Bay, Kowloon, Hong Kong. [2]Division of Life Science, Hong Kong University of Science and Technology, Clear Water Bay, Kowloon, Hong Kong. [3]State Key Laboratory of Nonlinear Mechanics, Institute of Mechanics, Chinese Academy of Sciences, Beijing, China. [4]School of Engineering Science, University of Chinese Academy of Sciences, Beijing, China. [5]School of Life Sciences, Southern University of Science and Technology, Shenzhen, China. [6]These authors contributed equally: Zhitao Liao, Bowen Jia. ✉e-mail: dsguan@imech.ac.cn; zhangmj@sustech.edu.cn; penger@ust.hk

Many recent studies of protein condensates have been focused on simple one- or two-component condensates[28–31], such as *C. elegans* protein PGL-3 and the human fused in sarcoma (FUS) protein[15,20,32]. These condensates have served as model systems for the study of the biochemical and fluid properties of the protein condensates. Because the molecular interactions in these systems are relatively simple, the Maxwell model was found to describe their mechanical behavior well[20,22]. A key feature of Maxwell fluids is that when a constant strain is applied, the resulting stress decays exponentially with a single relaxation time[33,34]. Biological condensates in living cells, however, generally contain multiple macromolecular components, such as proteins, nucleic acids and lipids, with complex multivalent interactions.

For many condensates in living cells, the LLPS is often driven by multivalent interactions arising from intrinsically disordered regions (IDRs), binding between folded domains and/or binding between folded domains and short binding motifs[1,3,35]. With increasing protein components and interaction complexities, emergent properties, such as forming percolated protein networks, are expected to arise in the protein condensates, leading to unusual mechanical behaviors beyond simple Maxwell fluids[5,36–39]. A quantitative description of the mechanical properties of functional condensates and a further understanding of their major roles in regulating essential cellular functions are, therefore, urgently needed.

Here, we report a systematic study of the emergent multi-scale mechanics and its molecular origin of a functional multi-component protein condensate using atomic force microscopy (AFM) based mesoscale rheology and quantitative fluorescence measurements of protein mobility and composition. The protein condensate chosen for the study is a reconstituted postsynaptic density (PSD) system, consisting of six essential postsynaptic proteins (6xPSD), PSD-95, GKAP, Shank3, Homer3, SynGAP and the NR2B tail (see Methods for details about sample preparation). The molecular origin of the six protein components is from synapse in central nervous system. One of the reasons for choosing the 6xPSD in this study is that the 6xPSD condensate mimics the actual functional excitatory PSDs in living neurons[5]. As illustrated in Fig. 1a, PSDs are subcellular and membraneless structures beneath the postsynaptic membrane that play an essential role in neuronal signal transduction[40]. Studying how the PSDs self-assemble and respond to mechanical cues is crucial for understanding synapse functions[41–44].

When the six protein components are mixed in a buffer solution at physiological concentrations, the protein solution undergoes phase separation through multivalent interactions and oligomerization[5]. Among the six protein components, PSD-95, GKAP, Shank3, and Homer3 are the core scaffold proteins that drive the phase separation and further recruit NR2B (glutamate receptor) and SynGAP (an enzyme). Though two protein components, such as PSD-95 and SynGAP, can induce phase separation at higher concentrations, the four scaffold proteins combined together are the main drivers for phase separation at physiologically relevant concentrations[45]. The 6xPSD condensate is a well-characterized system, and its biochemical properties have been fully documented in recent studies[5,46–49]. The six protein components in the 6xPSD system have been shown to play essential roles in building the PSD structures and maintaining the integrity and biological functions of neuronal synapses[5,46]. Therefore, the 6xPSD is an ideal condensate system for the study attempted here to provide a quantitative description of the mechanical properties of a functional multivalent protein condensate, leading to a further understanding of PSD formation and plasticity in living neurons.

In this work, we show that the majority (80%) of the proteins in the 6xPSD droplet are mobile and diffuse in a partitioned space made by a dynamically cross-linked network, consisting of the remaining (20%) non-mobile scaffold proteins. The binding/unbinding dynamics of the weak bonds in the protein network give rise to a microscopic time $\tau_{off}$ that determines the long-time behavior of the mechanical relaxation of the 6xPSD against mechanical perturbations. Specifically, the relaxation modulus $E(t)$ of the 6xPSD is found to have a power-law relaxation mode over a broad range of times $t$ ($>\tau_{off}$) with a power-law exponent close to 1/2. Our coarse-grained mechanical profiling together with the fluorescence measurements, thus provides a reliable experimental framework that can be utilized to quantitatively characterize the mechanical state of the protein condensates and investigate their physiological functions and connections with diseases.

## Results

### Force relaxation measurement reveals a two-mode relaxation modulus $E(t)$

We first conduct force (or stress) relaxation measurements on a 6xPSD droplet using an AFM and a colloidal probe of diameter ~15 μm, as sketched in Fig. 1b. The phase-separated droplets have a typical size $s \simeq$ 10–20 μm and height $h \simeq 2$ μm, as shown in Fig. 1c. The AFM probe is brought into contact with the droplet at a high loading speed (~100 μm/s, (i) approach) and then held at a fixed indentation $\delta$ (0.1–0.2 μm, (ii) hold) onto the droplet surface once the droplet restoring force reaches a preset maximal value $F_0$ (1–3 nN), as illustrated in Fig. 1d. Under this AFM operation, an almost instantaneous (~1 ms) and constant indentation $\delta$ is imposed onto the droplet surface, as illustrated in the bottom panel of Fig. 1d and as shown in Fig. 1e. In the meanwhile, the AFM records the resulting force evolution $F(t)$ over a 5-decade time span from 0.1 ms to 10 s.

During the hold stage, the contact area between the probe and droplet does not change much (see Supplementary Figs. 9–11, Supplementary Tables 1, 2, and Supplementary Section II.C for details), but the measured force $F(t)$ decays with time $t$. The normalized resulting force $F(t)/F_0$ averaged from 20 6xPSD droplets (black dots in Fig. 1f) clearly reveals two distinct relaxation regimes: a rapid exponential-like decay when time $t$ is less than the crossover time $t_c$ ($\simeq 9$ ms) and a slow power-law decay for $t > t_c$. The power-law relaxation lasts over a 3-decade time span (up to 10 s), until the measured force reaches its noise level (~0.02 nN) resulting from hydrodynamic and thermal fluctuations in the fluid (see Methods for more experimental details).

It is seen from Fig. 1f that the force relaxation for the 6xPSD condensates can be well described by

$$F(t)/F_0 = C_1 e^{-t/\tau_1} + C_2 \left(1 + t/\tau_2\right)^{-\alpha}, \qquad (1)$$

where $C_1$ and $C_2$ are the weighting factors of each relaxation mode, $\tau_1$ and $\tau_2$ are the corresponding relaxation times, and $\alpha$ is the power-law exponent. The best fit of Eq. (1) to the data (light grey solid line) gives $C_1 = 0.64$, $C_2 = 0.36$, $\tau_1 = 2.9$ ms, $\tau_2 = 4.9$ ms and $\alpha = 0.54$. Under the normalization condition, we have $C_1 + C_2 = 1$. Equation (1) thus provides a quantitative description of the multi-scale stress relaxation of the 6xPSD condensates. The two relaxation modes have a comparable weighting factor $C_i$, indicating that they contribute to the modulus of the 6xPSD condensates with a similar magnitude.

Using the linear relation $\sigma(t) = \varepsilon E(t)$ for a constant strain $\varepsilon \simeq \delta/h$ (with $\sigma(t)$ being the resulting stress), we find the time-dependent Young's modulus under compression (i.e., compressive modulus) $E(t)$ of the 6xPSD condensates (see Supplementary Section II.A.1 for details)

$$E(t) = E_1 e^{-t/\tau_1} + E_2 \left(1 + t/\tau_2\right)^{-\alpha}, \qquad (2)$$

where $E_1$ ($= E_0 C_1$) and $E_2$ ($= E_0 C_2$) are the component moduli of each relaxation mode, and $E_0$ ($= E_1 + E_2$) is the initial value of the total modulus at $t = 0$.

### Force-indentation relation $F(\delta)$ provides a consistent and quantitative mechanical profiling of the 6xPSD condensate

To determine the absolute value of $E_0$ ($= E_1 + E_2$), we conduct force-indentation measurement by keeping the colloidal probe

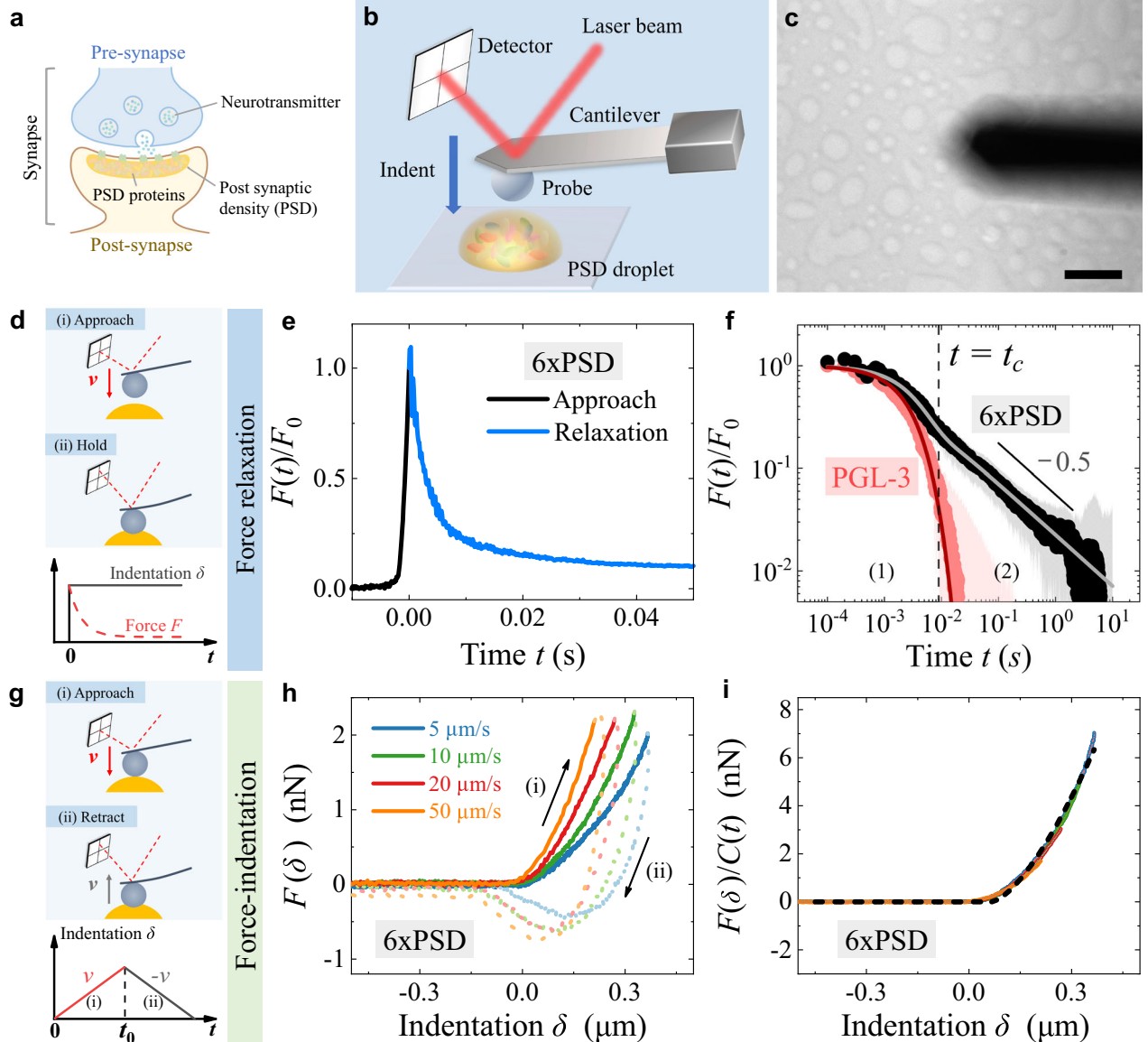

**Fig. 1 | Measurements of force relaxation and force-indentation relation reveal a two-mode relaxation modulus $E(t)$ for 6xPSD. a** Illustration of a neuronal synapse and postsynaptic density (PSD). **b** AFM setup for mechanical measurement of a 6xPSD droplet using a colloidal probe. **c** Bright-field image (top view) of the rectangular cantilever (dark finger) and 6xPSD condensates (bright droplets) on a coverslip. Scale bar: 30 μm. Similar images were obtained in multiple measurements ($N > 5$). **d** Illustration of AFM operation for force relaxation measurement (upper two panels) and a sketch of the resulting force $F(t)$ and indentation $\delta(t)$ as a function of time $t$ (bottom panel). **e** Linear plot of the normalized droplet restoring force $F(t)/F_0$, when the AFM probe approaches the droplet surface (black line with $t < 0$) and holds at a constant indentation $\delta$ (blue line with $t > 0$). **f** Log-log plots of the force relaxation curve $F(t)/F_0$ ($t > 0$) averaged over 20 6xPSD droplets (black dots) and over 10 PGL-3 droplets (red dots). The shaded areas indicate the standard deviations of the force measurements. The vertical dashed line indicates the crossover time $t_c$ ($\simeq 9$ ms) from the exponential decay regime (1) to the power-law decay regime (2) for 6xPSD. The black solid line indicates the power-law decay $\propto t^{-0.5}$. The light grey and dark red solid lines show the fits of Eq. (1) to the black dots and red dots, respectively. **g** Illustration of AFM operation for force-indentation measurement (upper two panels) and a sketch of the imposed indentation $\delta(t)$ as a function of time $t$ (bottom panel). **h** Measured force-indentation curves $F(\delta)$ for 6xPSD, when the AFM probe (i) approaches (colored solid lines) and (ii) retracts (colored dotted lines) from the droplet surface. The measurements are made at four loading speeds $v$ as indicated. **i** Normalized force-indentation curves $F(\delta)/C(t)$ during probe approaching with different loading speeds $v$. The data are taken from (**h**) with the same color codes. The black dashed line is a fit of Eq. (2) to the data with a single fitting parameter $E_0 = 9584$ Pa. Illustrative figures in (**a**, **b**) were partially adapted from refs. [45],[54]. Source data are provided as a Source Data file.

continuously compressing a 6xPSD droplet at a fixed speed $v$ ((i) approach) until the resulting force reaches a preset value $F_0$. The colloidal probe then retracts from the droplet at the same speed without halting, until it detaches from the droplet surface ((ii) retract), as illustrated in Fig. 1g. Figure 1h shows the resulting force $F(\delta)$ as a function of indentation $\delta$ for a single droplet at four different loading speeds. The measured approaching curves of $F(\delta)$ (colored solid lines) appear to be of Hertzian type (see

Supplementary Fig. 12) and reveal a strong dependence on the loading speed $v$. The droplet appears to be stiffer at a higher speed as it requires a larger force to achieve the same indentation. The measured retracting curves of $F(\delta)$ (colored dotted lines) reveal a strong speed-dependence and a large hysteresis between the approaching and retracting curves (see Supplementary Fig. 15 and Supplementary Section II.D.2 for details). The negative sign of the measured $F(\delta)$ before detachment indicates

**Table 1 | Measured mechanical properties of the 6xPSD condensate**

| Sample | $C_1$ | $C_2$ | $\tau_1$ (ms) | $\tau_2$ (ms) | $\alpha$ | $E_0$ (Pa) | $E_1$ (Pa) | $E_2$ (Pa) | $N$ |
|---|---|---|---|---|---|---|---|---|---|
| S1 | 0.56 ± 0.13 | 0.44 ± 0.13 | 2.8 ± 0.5 | 3.6 ± 1.7 | 0.54 ± 0.08 | 6999 ± 2071 | 3913 ± 1405 | 3077 ± 1385 | 20 |
| S2 | 0.54 ± 0.03 | 0.46 ± 0.03 | 5.9 ± 0.7 | 10.8 ± 2.0 | 0.50 ± 0.02 | 5211 ± 2443 | 2710 ± 1271 | 2501 ± 1173 | 2 |

The values of the weighting factors $C_1$, $C_2$, relaxation times $\tau_1$, $\tau_2$ and power-law exponent $\alpha$ are obtained from the stress relaxation measurements. The values of the initial modulus $E_0$ are obtained from the force-indentation measurements. The number $N$ indicates the number of droplets used in sample averaging. The error bars show the standard deviations (for $N = 20$) or fitting uncertainties (for $N = 2$) of the fitting results.

that there is an adhesion between the colloidal probe and the 6xPSD droplet at the late stage of surface contact.

We now show that the speed-dependence of $F(\delta)$ in the approaching direction is a key feature of viscoelasticity of the 6xPSD condensates. For a colloidal probe in contact with a pure elastic surface without adhesion, the force-indentation curve $F(\delta)$ is described by the Hertz model[50], $F(\delta) = 4ER^{\frac{1}{2}}\delta^{\frac{3}{2}}/[3(1 - \nu^2)]$, where $R = 1/(1/R_1 + 1/R_2)$ is the effective radius with $R_1$ and $R_2$ being the radii of the probe and droplet, $\nu$ is the Poisson ratio, and $E$ is the Young's modulus. The Hertz equation, which predicts that $F(\delta)$ is a single-valued function of $\delta$ and bears no speed-dependence and hysteresis, has been used widely to determine the elastic modulus $E$ of living cells and biological materials (with $\nu \simeq 0.5$)[5,51,52].

With the time-dependent modulus $E(t)$ in Eq. (2) (instead of a constant modulus $E$), we modify the Hertz model by using Ting's theory[53–55] and obtain

$$F(\delta, t) \simeq \frac{16}{9}E_0 R^{\frac{1}{2}}\delta^{\frac{3}{2}}C(t), \qquad (3)$$

where $C(t)$ is a correction factor and has the form

$$C(t) \simeq \frac{3}{2}C_1\left[\frac{\tau_1}{t} - \frac{\sqrt{\pi}}{2}\left(\frac{\tau_1}{t}\right)^{\frac{3}{2}}e^{-\frac{t}{\tau_1}}\text{Erfi}\left(\sqrt{\frac{t}{\tau_1}}\right)\right]$$
$$+ C_2\frac{3\sqrt{\pi}\Gamma(1-\alpha)}{4\Gamma\left(\frac{5}{2}-\alpha\right)}\left(\frac{t}{\tau_2}\right)^{-\alpha}. \qquad (4)$$

The value of $C(t)$ is determined by the time $t = \delta/v$ and the five relaxation parameters $C_1$, $C_2$, $\tau_1$, $\tau_2$, and $\alpha$ in Eq. (1) (see Supplementary Section II.A.1 for details).

As shown in Fig. 1i, all of the approaching curves of $F(\delta)$ obtained at different loading speeds $v$ collapse onto a master curve, once $F(\delta)$ is normalized by $C(t)$. The resulting master curve is well described by Eq. (3) with a single fitting parameter $E_0 = 9584$ Pa (black dashed line). The normalized plot of $F(\delta)/C(t)$ removes the speed-dependence in the measured $F(\delta)$ and recovers to the Hertzian form. Figure 1i thus demonstrates that the speed-dependence of the measured $F(\delta)$ is caused by the relaxation modulus $E(t)$, which explains why the measured curves and contact area are Hertzian type (see Supplementary Figs. 9–11 and Supplementary Section II.C for details). The two-mode relaxation of the measured $E(t)$ is also verified by other force relaxation measurements using different loading protocols (see Supplementary Fig. 7 and Supplementary Section II.A.1 for details). The surface adhesion during the probe retraction, as shown in Fig. 1h, turns out not playing an important role in the measurements of force relaxation and approaching curves of the measured $F(\delta)$ (see Supplementary Figs. 13–14 and Supplementary Section II.D.1 for details).

The two types of AFM measurements of stress relaxation and force-indentation relation complement each other, and they jointly provide a consistent and quantitative description of the viscoelastic properties of the 6xPSD condensate. Table 1 gives a summary of the fitting results of the multi-scale mechanical properties of the 6xPSD condensates. The values of $E_0$ used in Table 1 are adjusted by the finite thickness correction due to the small height of the 6xPSD droplets (see Supplementary Section II.A.2 for details). In addition to Sample S1 used

in Fig. 1, we also analyze the data from Sample S2 obtained in our previous study[5]. It is found that Sample S2 exhibits the same two-mode relaxation as Sample S1 does (see Supplementary Fig. 17 and Supplementary Section II.E for details). There are some slight differences in the fitting parameters, as shown in Table 1, which might be caused by the variations between different sample batches and slow aging effect of Sample S2 due to different measurement times. Nonetheless, the two sets of data provide a consistent multi-scale mechanical profiling of the 6xPSD condensates.

**Complex shear modulus reveals non-Maxwell behaviors of the 6xPSD condensate**

The relaxation modulus $E(t)$ is commonly used to describe the viscoelasticity of polymeric fluids[33,34]. Its Fourier transform is directly linked to the complex shear modulus $G^*(\omega)$, which is an alternative way of characterizing viscoelastic materials in the frequency domain (see Supplementary Section II.B.1 for details). Since the two relaxation modes in Eq. (2) are well separated in time, using $E(t)$ is more convenient for identifying the molecular origins of the condensate's viscoelasticity.

For comparison, we perform the same relaxation measurements for a model single-component protein condensate, PGL-3, which behaves as a Maxwell fluid[20,56,57]. As shown in Fig. 1f, the measured force relaxation $F(t)/F_0$ for the PGL-3 droplets (red dots) decays to zero much faster than that for 6xPSD and can be well described by a simple exponential decay without power-law relaxation. The best fit of Eq. (1) to the data (dark red solid line) gives $C_1 = 1$, $\tau_1 = 2.8$ ms, and $C_2 = 0$. The single exponential decay in the relaxation modulus $E(t)$ is a hallmark of Maxwell fluids[33,34]. The results shown in Fig. 1f thus confirm that the PGL-3 condensate is of Maxwell type, as previously reported.

From the measured $E(t)$, we obtain the complex shear modulus $G^*(\omega)$ for PGL-3 (see Supplementary Section II.B.1 for details). As shown in Fig. 2a, the obtained $G'(\omega)$ and $G''(\omega)$ for PGL-3 are well described by the Maxwell model (grey and red dashed lines) and scale as $G'(\omega) \sim \omega^2$ and $G''(\omega) \sim \omega$ at low frequencies, which are the key features of Maxwell fluids. The obtained results for PGL-3 are in good agreement with the previous experiments[20,56] (see Supplementary Section II.B.1 for details). In contrast to PGL-3, the 6xPSD condensate exhibits non-Maxwell behaviors, as shown in Fig. 2b. First, the measured $G'(\omega)$ for 6xPSD has an overall amplitude ~10 times larger than that for PGL-3, indicating that the 6xPSD condensate has a much larger elastic modulus. Second, the measured $G'(\omega)$ and $G''(\omega)$ for 6xPSD show significant deviations from the Maxwell model in the low-frequency regime. Both the elastic and viscous moduli for 6xPSD decrease with decreasing $\omega$ much slower than the prediction for Maxwell fluids. This observation is consistent with the stress relaxation measurements (see Fig. 1f), which shows that the relaxation modulus for 6xPSD has a slow power-law relaxation $E(t) \sim t^{-0.5}$ at long times, corresponding to a power law $\propto \omega^{0.5}$ in the frequency domain, as marked in Fig. 2b (see Supplementary Section II.B.1 for details). Figures 1f, 2a, b thus demonstrate that our AFM methodology of measuring the relaxation modulus $E(t)$ works, and the power-law relaxation in the measured $F(t)/F_0$ is an emergent property of 6xPSD.

We also performed droplet coalescence experiments to examine the interfacial properties of the 6xPSD condensate. By putting two 6xPSD droplets in contact, the droplets gradually merge into a single

**Fig. 2 | Comparison of complex shear modulus $G^*(\omega) = G' + iG''$ between the PGL-3 and 6xPSD condensates and droplet coalescence measurements.**
**a**, **b** Obtained storage modulus $G'(\omega)$ (black squares) and loss modulus $G''(\omega)$ (red circles) as a function of angular frequency $\omega$ for the PGL-3 (**a**) and 6xPSD (**b**) condensates. The grey and red dashed lines in (**a**, **b**) show the Maxwell fits to the data points. The black solid lines indicate the power laws: $\omega^1$, $\omega^2$, and $\omega^{0.5}$ in the low-frequency regime. The data represent an average over 20 6xPSD droplets and 10 PGL-3 droplets, respectively. The error bars show the standard deviation. **c** Time lapse images showing the merging of two 6xPSD condensate droplets into a single spherical droplet under the action of interfacial tension $\gamma$, as shown in Fig. 2c. Figure 2d shows the time evolution of droplet aspect ratio AR($t$) during the coalescence. The measured AR($t$) is well described by an exponential decay function, $\mathrm{AR}(t) = 1 + A_0 \exp(-t/\tau_a)$, where $\tau_a$ is the characteristic coalescence time. The value of $\tau_a$ is determined by[6,7,9,22,58], $\tau_a \simeq R_2 \eta/\gamma$, where $\eta$ is the droplet viscosity and $R_2$ is its radius. Figure 2e summarizes the resulting values of $\eta/\gamma = 12.1 \pm 7.6$ s/μm from 33 6xPSD droplets.

spherical droplet. Scale bar: 15 μm. Similar images were obtained in multiple measurements ($N = 33$). **d** Time-dependence of the measured aspect ratio AR($t$) of the merging droplet in (**c**) (black dots). The red solid line shows a fit to the exponential decay function, $\mathrm{AR}(t) = 1 + A_0 \exp(-t/\tau_a)$, with amplitude $A_0 = 1.1$ and coalescence time $\tau_a = 96$ s. **e** Obtained values of $\eta/\gamma$ (black diamonds) from multiple 6xPSD droplets ($N = 33$). The light blue box shows the mean value ± standard deviation of the measurements ($\eta/\gamma = 12.1 \pm 7.6$ s/μm). Source data are provided as a Source Data file.

Using the obtained $G^*(\omega)$ for the 6xPSD droplets, we estimated their effective viscosity $\eta \simeq 2.1$ Pa · s, which is about 14 times larger than that of the fresh PGL-3 condensate (see Supplementary Section II.B.1 for details). We then obtain the interfacial tension of the 6xPSD droplets in water, $\gamma \simeq 0.2$ μN/m, which is of the same order as that for PGL-3[20]. Because the interfacial tension of the 6xPSD droplets is very small, the capillary force associated with the droplet-buffer interface is negligibly small compared to $F_0$ (see Methods for details).

## Quantitative fluorescence measurements reveal protein mobility and concentration distributions in the 6xPSD condensate

To link the mechanical properties of the 6xPSD condensates to their molecular structures, we conduct additional (separate) measurements of fluorescence recovery after photobleaching (FRAP) in the condensed phase. By labeling each protein component of the 6xPSD condensates one at a time with a fluorescent dye Cy3, the FRAP data allows us to examine the protein mobility inside the 6xPSD droplets (Fig. 3a–c). In particular, the FRAP can quantify the two-dimensional (2D) lateral diffusion of the labeled protein molecules in the condensed phase. For a cylindrical laser bleaching column of radius $r$, the 2D lateral diffusion gives rise to an FRAP intensity profile[59,60]

$$I(t) = Ae^{-\tau/(2t)}\left[I_0\left(\frac{\tau}{2t}\right) + I_1\left(\frac{\tau}{2t}\right)\right], \qquad (5)$$

where $I_0$ and $I_1$ are, respectively, the zeroth- and first-order modified Bessel function, $\tau = r^2/D$ is the FRAP recovery time by protein diffusion with $D$ being the diffusion coefficient, and $A$ is the saturation value (see Methods for details).

It is seen from Fig. 3c that the FRAP mean intensity profiles $I(t)$ for the six protein components can all be well described by Eq. (5) with only two fitting parameters, $A$ and $D$ (the value of $r$ is known). When the FRAP data are plotted as $I(t)/A$ versus $t/\tau$, the six different intensity profiles in Fig. 3c collapse on a single master curve (see Supplementary Fig. 1), further confirming that the six different protein components in the condensed phase share the same diffusion mechanism (but with

different values of $A$ and $D$). The fitted values of $D$ for each protein component are given in Table 2. The fitted values of $A$ and $D$ are barely affected by the bleaching area selected in the experiment, as they fall within the quoted error bars (see Supplementary Fig. 3 and Supplementary Section I.A. for details. The values of $A$ and $1 - A$ tell us, respectively, what fraction of the proteins is mobile and what fraction of the proteins is immobile at time $t \to \infty$, from which we obtain the mobile/immobile fractions of each protein components in the condensed phase at a finite cut-off time $t/\tau = 3$ (typically 6–60 min in real time depending on the type of proteins, see Supplementary Section I.A. for details).

For example, Table 2 reveals that each protein component has a fraction of proteins (in the range of (8–45)%), which are held immobile by forming a stable protein network in the condensed phase (see more discussions below), and thus they cannot be replaced by non-bleached protein molecules for fluorescence recovery. To verify that the mobile/immobile fractions of proteins indeed form two stable interpenetrating components of the condensed phase with distinct mechanical and fluorescent relaxations, we repeat photo-bleaching and recovery measurements three times in a row on the same droplet, and each measurement runs over a 10-minute interval. It is found that the resulting FRAP intensity profiles $I(t)$ in the second and third measurements fully reproduce the results in the first measurement (see Supplementary Fig. 2). This result suggests that the mobile/immobile fractions of proteins remain stable over a long period (~30 min.).

We also use a confocal-imaging-based technique of fluorescence intensity quantification (FIQ) to determine the absolute concentration of each protein component in the condensed phase. This method has been fully characterized in recent experiments[5,61,62] (see Supplementary Fig. 4 and Supplementary Section I.B for details). With a 3D confocal scanning, we locate the brightest horizontal cross-section (XY plane) of a single droplet and the corresponding intensity profile, as shown in Fig. 3d. When compared with a calibrated fluorescence intensity versus Cy3 concentration curve, as illustrated in Fig. 3e, we obtain the absolute concentration of each protein component in the condensed phase (see Table 2). The obtained concentration ratios between different protein components are further calibrated by a centrifuge assay, as shown in Fig. 3f, g and Supplementary Fig. 5 (see Methods and Supplementary Section I.C. for details). The band intensities in the gel are used to quantify the pellet (condensed phase) percentage $p$ and supernatant (dilute phase) percentage $1 - p$ of each protein component. As shown in Table 2, the condensed phase exhibits a protein enrichment of approximately 8 mM/30 μM $\simeq$ 260 folds compared to the dilute

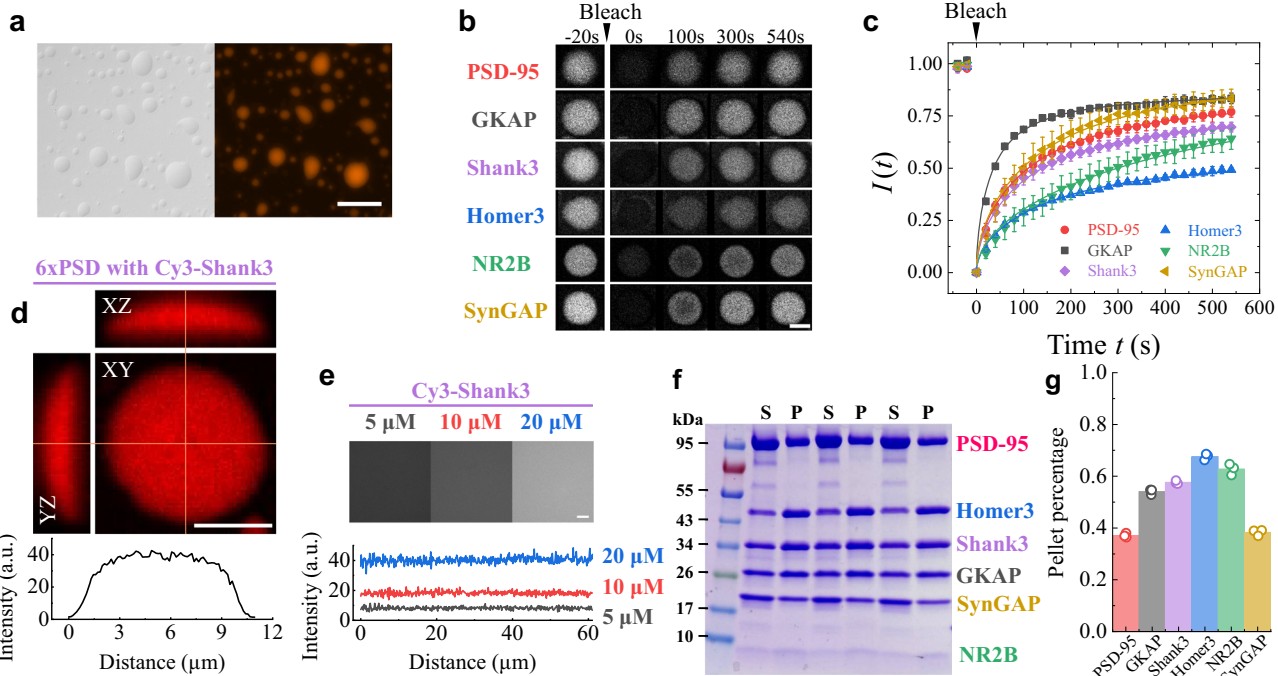

**Fig. 3 | Quantitative fluorescence measurements reveal the mobile/immobile partition of proteins in the 6xPSD condensate. a** Images of differential interference contrast (DIC, left) and fluorescence (right) of the 6xPSD droplets with 1% PSD-95 labeled by a fluorescent dye Cy3. Scale bar: 20 μm. **b** Six example sets of time-lapse images of FRAP obtained from individual 6xPSD droplets with 1% single protein component being labeled by Cy3. Scale bar: 5 μm. **c** Normalized FRAP mean intensity $I(t)$ of the 6xPSD droplets as a function of time $t$ before ($t < 0$) and after ($t > 0$) photo-bleaching. The mean value $I(t)$ is obtained by averaging the FRAP intensity over 8 droplets for PSD-95, 4 droplets for GKAP, 9 droplets for Shank3, 8 droplets for Homer3, 8 droplets for NR2B and 10 droplets for SynGAP. The error bars show the standard deviations of the measurements. The solid lines are the best fits of Eq. (5) to the FRAP data. **d** Two-dimensional (2D) fluorescence intensity maps obtained by 3D confocal scanning of a 6xPSD droplet with 1% Shank3 labeled by Cy3 (upper panel). Scale bar: 5 μm. A 1D intensity profile across the diameter of a

horizontal section (XY plane) of the droplet at its brightest section (bottom panel). a.u. arbitrary units. Similar images were obtained in multiple experiments ($N \geq 3$). **e** Fluorescence intensity images obtained from a solution of the protein Shank3 labeled by Cy3 with three different concentrations: 5 μM, 10 μM, and 20 μM (upper panel, scale bar: 5 μm) and the corresponding 1D intensity profiles (bottom panel). a.u. arbitrary units. Similar images were obtained in multiple experiments ($N \geq 3$). **f** Three repeated runs of the centrifuge assay[5], in which a high-speed centrifuge is used to separate the dilute phase (Supernatant, S) and condensed phase (Pellet, P). Protein components in each phase are further separated by electrophoresis (SDS-PAGE). The band intensities in the gel are used to quantify the pellet percentage and supernatant percentage of each protein component (the sum of the two percentages is unity). The NR2B band is quantified by the gel with sliver staining (see Supplementary Fig. 5). **g** Obtained pellet percentage of the six protein components from $N = 3$ measurements. Source data are provided as a Source Data file.

## Table 2 | Measured protein compositions and their mobile/immobile fractions and diffusivity in the 6xPSD condensate

| Protein components | PSD-95 | GKAP | Shank3 | Homer3 | NR2B | SynGAP | Total |
|---|---|---|---|---|---|---|---|
| Conc. before LLPS (μM) | 5 | 5 | 5 | 5 | 5 | 5 | 30 |
| Mobile fraction | (82 ± 8)% | (80 ± 10)% | (75 ± 6)% | (55 ± 5)% | (92 ± 8)% | (92 ± 8)% | 80% |
| Immobile fraction | 18% | 20% | 25% | 45% | 8% | 8% | 20% |
| $D$ (μm²/s) | 0.037 ± 0.015 | 0.063 ± 0.017 | 0.034 ± 0.007 | 0.026 ± 0.008 | 0.0087 ± 0.0022 | 0.021 ± 0.010 | 0.032 |
| Conc. after LLPS (mM) | 0.93 ± 0.09 | 1.61 ± 0.22 | 1.58 ± 0.09 | 1.40 ± 0.45 | 1.31 ± 0.41 | 1.14 ± 0.09 | 7.97 |
| Mobile conc. $n_{mo}$ (mM) | 0.76 | 1.29 | 1.18 | 0.77 | 1.21 | 1.05 | 6.26 |
| Immobile conc. $n_{im}$ (mM) | 0.17 | 0.32 | 0.40 | 0.63 | 0.10 | 0.09 | 1.71 |

The master protein solution before liquid-liquid phase separation (LLPS) has six protein components with equal concentrations of 5 μM each. The mobile/immobile fractions and the diffusion coefficient $D$ of each protein component in the condensed phase are obtained from the FRAP data (Fig. 3c) with the fitting to Eq. (5). The value of $D$ in the last column is the concentration-weighted average of the six component diffusion coefficients. The absolute concentration of Shank3 is determined solely by FIQ. The absolute concentration of the other five protein components is determined by taking the mean value from the measurements of FIQ and the centrifuge assay (see Table 3 and Methods for details). Combining the techniques of FRAP, FIQ and centrifuge assay, we obtain the concentration distribution of mobile/immobile proteins and their diffusion coefficients among the six protein components in the condensed phase. The error bars quoted in Table 2 are the standard deviations of the FRAP measurements obtained from 4 to 10 6xPSD droplets.

phase, with the concentration of each protein components at ~1 mM. From the obtained protein concentration and mobile/immobile fractions, we obtain the mobile/immobile protein concentrations of each protein component in the condensed phase.

## Discussion

From the above fluorescence measurements, we find two mechanically distinct and spatially intertwined structure components in the condensed phase. The minority component (20%) is made of

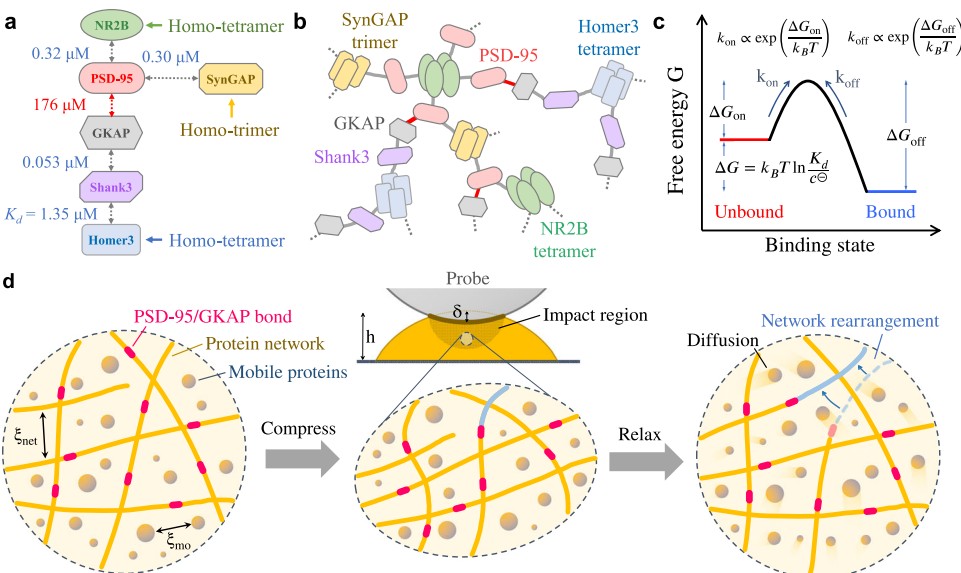

**Fig. 4 | Illustration of multivalent interactions, protein oligomerization, and network formation in the 6xPSD condensate. a** Specific multivalent interactions among the six components of proteins in the condensed phase and their dissociation constants $K_d$ (dashed double arrows). The self-oligomerization of proteins allows SynGAP to form trimers and NR2B and Homer3 to form tetramers (solid arrows). More details about the protein-protein interactions can be found in refs. 5,45,77–80. **b** A sketch of a protein network formed by the immobile proteins in the condensed phase. This large protein network has weak bonds (marked in red), which are formed by the PSD-95/GKAP interactions. **c** Free energy change and reaction constants associated with the binding and unbinding of the PSD-95/GKAP proteins. This reaction is assumed to be of the form $A + B \rightleftharpoons AB$, with the free energy change, binding affinity and on-and-off rates as marked. **d** Schematic diagrams showing how the 6xPSD condensate under compression relaxes mechanically. Before the loading of the AFM probe, the protein network with mesh size $\xi_{\text{net}}$ and

mobile proteins with separation $\xi_{\text{mo}}$ are both at equilibrium, and their structures remain isotropic (left panel). The weakest PSD-95/GKAP bonds in the network are marked in red. When a 6xPSD droplet of height $h$ is subjected to a sudden indentation $\delta$ (upper middle panel), the shaded impact region is compressed vertically by the colloidal probe, and the protein condensate becomes anisotropic (lower middle panel). The protein condensate in this non-equilibrium squeezed state will relax back to equilibrium (right panel) via a fast exponential relaxation of the disturbed concentration variations of mobile proteins by diffusion followed by a slow power-law relaxation of the deformed protein network at length scales larger than $\xi_{\text{net}}$. The network relaxation (or rearrangement) is regulated by the binding and unbinding of the (weakest) PSD-95/GKAP bonds in the network. As illustrated in the right panel, a network filament (blue solid line) binds to a new site after the PSD-95/GKAP bond unbinds from its original position (blue dashed line). Illustrative figures in (**d**) were partially adapted from ref. 54.

immobile protein molecules, and they form a relatively stable network via multivalent interactions. This finding is also supported by a recent single molecular tracking study of a similar reconstituted PSD[38], which revealed that the protein diffusion inside the condensed phase is hindered significantly and becomes spatially heterogeneous. Given the total immobile protein concentration $n_{\text{im}} = 1.71$ mM (see Table 2), the immobile protein network will have a typical mesh size $\xi_{\text{net}} \simeq 2\sqrt[3]{3/(4\pi N_A n_{\text{im}})} = 12.3$ nm, where $N_A$ is Avogadro's constant (see Supplementary Section II.A.3 for details). This value of $\xi_{\text{net}}$ is consistent with the estimated mesh size of other protein condensates[63–65]. Similarly, for the mobile proteins in the condensed phase with the concentration $n_{\text{mo}} = 6.26$ mM (see Table 2), we find the typical size of the cage formed by the surrounding molecules, $\xi_{\text{mo}} \simeq 2\sqrt[3]{3/(4\pi N_A n_{\text{mo}})} = 8.0$ nm, which is 65% smaller than $\xi_{\text{net}}$.

The majority component (80%) is made of mobile protein molecules and they have a molecular weight ranging from 5 kDa to 95 kDa (see Fig. 3f). For the largest molecular weight of 95 kDa, we find the corresponding molecular size (or radius of gyration) $R_g \simeq 2.9$ nm[66–68] and overlap concentration[69,70] $n^* \simeq 3/(4\pi N_A R_g^3) = 16$ mM. Because the total mobile protein concentration in the condensed phase, $n_{\text{mo}} = 6.26$ mM is smaller than $n^*$, the mobile protein molecules (and their oligomers) are in the dilute regime so that they can diffuse through the immobile protein network without much influence from the neighboring mobile proteins. Consequently, their FRAP recovery curves follow the same diffusion mechanism. In the following, we demonstrate that these two structure components are responsible for the two-mode viscoelastic relaxation, as shown in Eq. (2).

## Exponential relaxation at short times indicates a diffusive relaxation of mobile proteins in the condensed phase

This relaxation results from the diffusive rearrangement of local concentration variations of mobile proteins induced by compression. The diffusive relaxation time goes as[33], $\tau_1 \simeq \xi_{\text{mo}}^2/D$, where the cage size $\xi_{\text{mo}}$ is used as the relevant length for concentration variations in the condensed phase and $D$ is the diffusion coefficient. As the mobile proteins have different binding rates to the protein network and different mutual interactions, the six protein components have different diffusion coefficients $D$ in the condensed phase (see Table 2). For example, GKAP diffuses much faster than the other five proteins, because its binding to PSD-95 is much weaker and thus its diffusion is limited primarily by the interactions with Shank3. The motion of the other five protein components, on the other hand, is hindered by self-oligomerization and more interacting partners (see Fig. 4a). Despite these complications, we use a concentration-weighted average value, $\langle D \rangle_c \simeq 0.032$ μm²/s, as a representative diffusion coefficient. With the measured values of $\langle D \rangle_c$ and $\xi_{\text{mo}}$, we find the relaxation time $\tau_1 \simeq \xi_{\text{mo}}^2/\langle D \rangle_c \simeq 2.0$ ms, which is close to the measured value $\tau_1 = 2.8$ ms. A similar result is also obtained for the PGL-3 droplets (see Supplementary Section II.B.2 for details).

Since the individual diffusing oligomers share the same thermal energy $k_B T$, the relaxation modulus $E_1$ should scale with the number density $n_{\text{oli}}$ of the diffusing protein oligomers, i.e., $E_1 \simeq N_A k_B T n_{\text{oli}}$[33,34]. From the measured $E_1 \simeq 3.9$ kPa (see Table 1), we find $n_{\text{oli}} \simeq 1.6$ mM, which is smaller than the total mobile protein concentration $n_{\text{mo}} = 6.26$ mM (see Table 2). This result suggests that the mobile proteins in the condensed phase form protein oligomers with an average oligomerization number of $n_{\text{mo}}/n_{\text{oli}} \simeq 3.9$. The above

calculations thus confirm that the short-time exponential decay is determined primarily by the local concentration rearrangement of mobile proteins in the (deformed) condensate.

### Power-law relaxation at long times reveals the cross-linking dynamics of the immobile protein network in the condensed phase

The power-law relaxation in the measured $E(t)$ is caused mainly by the slow relaxation of the deformed immobile protein network itself. In the time domain, the power-law decay is described by $E_2(1 + t/\tau_2)^{-\alpha}$, where the constant term of unity is introduced to separate the power-law relaxation from the short-time exponential relaxation. As shown in Fig. 1f, the two relaxation modes are well separated by a crossover time $t_c \simeq 2.5\tau_2$ (~9 ms), after which $E(t > t_c) \simeq E_2(t/\tau_2)^{-\alpha}$ for $t/\tau_2 \gg 1$. Physically, $\tau_2$ represents the fastest relaxation associated with the smallest scale of the protein network.

Power-law rheology with $0 \le \alpha \le 1$ is used to characterize the slow reorganization of soft glassy networks, which have a broad spectrum of relaxation times because of structural disorder and metastability[71]. The power-law exponent $\alpha$ has been systematically evaluated in the range of 0.1–0.5 for living cells and other soft glassy materials[54,72]. In the frequency domain, the power-law rheology implies both the storage modulus $G'(\omega)$ and loss modulus $G''(\omega)$ have the same power-law scaling, $(\omega\tau_2)^{\alpha}$, at the low-frequency limit (or long-time limit), and their amplitude ratio varies with $\alpha$ as $G''(\omega)/G'(\omega) = \tan(\frac{\pi}{2}\alpha)$ (see Supplementary Section II.B.1 for details). In this case, $\alpha = 1/2$ becomes a special crossover value, below which the network behaves more like a solid (i.e., $G''(\omega)/G'(\omega) < 1$) and above which the network behaves more like a fluid (i.e., $G''(\omega)/G'(\omega) > 1$).

A key feature that distinguishes the protein networks from other polymeric networks, such as strong hydrogels and rubber-like materials, is the transient nature of their cross-links, which gives rise to an unusual stress relaxation that allows the network to flow at long times[73–76]. In particular, the model of cross-link-governed dynamics (CGD)[75] provided a mechanical description to this dynamic behavior and made two specific predictions. First, the stress relaxation of biopolymer networks is determined by independent unbinding activities of cross-links, which leads to a power-law relaxation with $\alpha = 1/2$. Second, the power-law relaxation starts at time $t \gtrsim \tau_{\text{off}}$, where $\tau_{\text{off}}$ is the characteristic time for a linker to unbind.

We now demonstrate that the measured power-law relaxation in the condensed phase is well described by the CGD model. First, as shown in Table 1, the obtained values of the power-law exponent $\alpha$ are very close to the predicted value of $\alpha = 1/2$. Second, the obtained power-law relaxation time $\tau_2$ is indeed associated with the unbinding time $\tau_{\text{off}}$ of a cross-linker, i.e., $\tau_2 = \tau_{\text{off}}$. Figure 4b shows a sketch of the protein network formed by the immobile proteins in the condensed phase. The interaction strength between the six proteins in the condensed phase is represented by their dissociation constants $K_d$ (see Fig. 4a), which were measured in previous experiments[5,45,77–80]. Among the specific protein interactions, the binding between PSD-95 and GKAP is probably the weakest one, as it has the largest dissociation constant $K_d = 176$ μM. For an interconnected network, as shown in Fig. 4b, its mechanical response to an external perturbation is determined primarily by the weak bonds of the network rather than its strong bonds, as the weak bonds unbind faster than the strong bonds[81,82].

In this case, the network starts to make rearrangements at its fastest unbinding time $\tau_{\text{off}} = 1/(k_{\text{on}}K_d) \simeq 5.6$ ms, where $k_{\text{on}} \simeq 10^6 \text{M}^{-1} \cdot \text{s}^{-1}$ is a typical binding rate for proteins[83,84]. The calculated protein unbinding time $\tau_{\text{off}}$ is very close to the measured power-law relaxation time $\tau_2$. On the other hand, the unbinding time for stronger protein bonds, such as that for GKAP/Shank3 and Shank3/Homer3, falls in the range of 1 s or even longer, in which most mechanical deformations of the network have relaxed through the PSD-95/GKAP unbinding.

This result thus explains why other cross-linkers in the network do not contribute much to the power-law relaxation of the 6xPSD condensates.

Because the power-law relaxation comes mainly from the binding and unbinding of the PSD-95/GKAP bonds, the component modulus $E_2$ should be linked to the energy density associated with the PSD-95/GKAP bonds that unbind from the network under compression, namely, $E_2 \simeq N_A(n_{\text{im}})_{95}E_b$, where $(n_{\text{im}})_{95} = 0.17$ mM is the molar density of the PSD-95/GKAP bonds (see Table 2) and $E_b$ is the binding energy of the PSD-95/GKAP bond. As shown in Fig. 4c, the free energy change of the PSD-95/GKAP is $\Delta G = k_B T \ln(K_d/c)$, where $c^{\ominus} = 1$ M is the concentration standard[83,84]. The energy required for the PSD-95/GKAP unbinding is $E_b = -\Delta G \simeq 8.6 k_B T$. With these numbers, we find $E_2 \simeq 3.6$ kPa, which is close to the measured value of $E_2$. The above analysis thus reveals an interesting dynamical structure of the immobile protein network in the 6xPSD condensates, whose mechanical property is determined primarily by the binding/unbinding dynamics of the PSD-95/GKAP bonds. This mechanism is further tested by introducing targeted modifications to the scaffold protein GKAP (see Supplementary Fig. 16 and Supplementary Section II.D.3 for details).

Unlike single-component protein condensates formed by intrinsically disordered regions (IDRs)[20,22], the 6xPSD condensate clearly exhibits a non-Maxwell behavior, i.e., power-law rheology resulting from the rearrangements of a percolated network inside the condensate. The cross-linked protein network has many routes for stress relaxation and thus gives rise to a broad range of relaxation times in response to external mechanical cues, in contrast to simple Maxwell fluids with a single relaxation time. In living cells, most of the biological condensates comprise multiple components and involve multivalent interactions arising from IDRs and/or specific bindings, which are the building blocks for making more complex network structures in the biological condensates than Maxwell fluids could do. Our model thus provides a further understanding of the multi-scale mechanics of functional protein condensates relevant to living cells.

Another possible relaxation channel for a networked medium is poroelastic relaxation due to the solvent movement out of the porous matrix in the compressed region of the network[85]. An important assumption that the poroelastic model made is that the porous matrix of the medium remains stable and does not relax during the poroelastic transport[86]. In other words, the lifetime of the protein network itself needs to be longer than the poroelastic relaxation time $\tau_p$. This may be true for certain polymeric networks, such as strong hydrogels and rubber-like materials, in which the lifetime of their cross-links is longer than $\tau_p$. The protein network in the 6xPSD condensate, on the other hand, is weak and dynamic compared with covalent bonds and its cross-links bind and unbind continuously over a range of times, which gives rise to a separate stress relaxation channel that allows the network to flow at long times[73–76]. As shown in Fig. 1f, the measured $F(t)/F_0$ (i.e., the accumulated stress) has decayed more than 95% through the power-law relaxation over a time span of 0.01–1 s, whereas the poroelastic relaxation has a typical relaxation time in the range of 1–10 s[54,85]. Therefore, the poroelastic model is not applicable to dynamic protein networks, such as that in the 6xPSD condensate, whose lifetime is shorter than the poroelastic relaxation time $\tau_p$ (see Supplementary Fig. 8 and Supplementary Section II.B.2 for details).

In conclusion, our AFM-based mescoscale rheology and fluorescence measurements establish that the 6xPSD condensate as a special type of viscoelastic material has a relaxation modulus $E(t)$, which is characterized by a set of five parameters ($E_1, \tau_1; E_2, \tau_2, \alpha$), as shown in Eq. (2). The interesting network structure of the 6xPSD allows the 6xPSD droplets to recruit and/or disperse certain proteins for molecular signaling and sustain mechanical cues through diffusive rearrangements in the short time and dynamic cross-link-governed network reorganization in the long time, as shown in Fig. 4d. The power-law rheology of the 6xPSD network is found to be determined

primarily by the binding/unbinding dynamics of the weakest PSD-95/GKAP bonds. This finding suggests an important route to alter the network structure of the 6xPSD droplets and hence regulate their physiological functions. For example, by modifying the specific interactions of the immobile scaffold proteins in the network via post-translational modification or mutation, one may change the binding affinity of the weak protein bonds sensitively and induce subsequent structural changes to study dendritic spine enlargement, which is relevant to synapse homeostasis and plasticity[41–43,47].

The ability of a condensate droplet to resist deformation, maintain structural integrity against entropic mixing, and transport functional proteins in and out of the droplet relies on an percolated network of scaffold proteins and regulatory proteins. At a more complex level, eukaryotic cells have a similar coarse-grained mechanical structure with mobile proteins in the cytosol partitioned by cytoskeletal networks. It was shown in a recent study[54] that the living cells have a similar relaxation modulus $E(t)$ with short-time exponential and long-time power-law relaxation. Because the typical mesh size of the protein network in the 6xPSD is similar to that of the cytoskeletal network, the exponential relaxation in the two systems has a similar diffusive relaxation time of 3–5 ms.

Compared to the living cells, we find the value of $E_0$ $(= E_1 + E_2)$ for the 6xPSD is approximately three times larger than that of living neurons[54], which suggests that the overall protein concentration in the 6xPSD droplets is larger than the surrounding cytosol. A higher protein density or larger viscoelastic modulus in the PSD condensates could help to provide a more stable synaptic connection between neurons. By using a quantitative approach that combines analytic mechanical profiling with molecular biological techniques, we obtain a unified description of the multi-scale mechanics of the 6xPSD and its molecular origin, which lay down a solid foundation for further studies on how the phase-separated PSD condensates regulate various biological functions and intracellular activities, in relation to their healthy and diseased states.

## Methods

### Protein purification, fluorescence labeling and sample preparation

All six post-synaptic density (PSD) proteins, PSD-95, GKAP, Shank3, Homer3, SynGAP, and NR2B, are expressed and purified as described in a previous publication[5]. The proteins are generated using standard PCR-based methods and cloned into vectors containing an N-terminal TRX-His$_6$/His$_6$-affinity tag followed by an HRV 3C cutting site. All constructs are confirmed by DNA sequencing. Recombinant proteins are expressed in *Escherichia coli* BL21 cells in LB medium at 16 °C overnight, and protein expression is induced by 0.25 mM IPTG (final concentration) at $OD_{600}$ between 0.6-0.8. Recombinant proteins are then purified using a Ni$^{2+}$-NTA agarose affinity column followed by size-exclusion chromatography (Superdex 200 or Superdex 75 as see fit) with a column buffer containing 50 mM Tris, 100 mM NaCl, 1 mM EDTA, and 1 mM DTT (pH 8.2). Recombinant protein tags are cleaved by HRV 3C protease and separated by another size-exclusion chromatography step. For PSD-95 and Homer3, a MonoQ column is applied to remove DNA contamination and is followed by a HiTrap desalting column to change to the buffer containing 50 mM Tris, 100 mM NaCl, 1 mM EDTA, and 1 mM DTT (pH 8.2).

To label proteins with a fluorescent tag, the purified proteins are exchanged into a HEPES buffer containing 20 mM HEPES, 100 mM NaCl, 1 mM EDTA and 1 mM DTT (pH 7.8) and concentrated to 5–10 mg/mL. Cy3 NHS ester dye (AAT Bioquest) is dissolved by DMSO and incubated with the corresponding protein at room temperature for 2 h (fluorophore to protein molar ratio is 1:1). Reaction is quenched by 200 mM Tris (pH 8.2). A HiTrap desalting column is applied to remove free dyes and change the labeled proteins into the buffer containing 50 mM Tris, 100 mM NaCl, 1 mM EDTA, and 1 mM DTT (pH

8.2). Both the labeled and unlabeled protein concentrations are calibrated using the NanoDrop One spectrophotometer (Thermo Fisher).

The recombinant PGL-3 proteins used in this study are expressed in *E. coli* BL21-CodonPlus (DE3) with His$_6$ tag. The protein is induced by 0.3 mM IPTG for 16 h at 18 °C and collected by centrifugation. After centrifugation for 15 min, the *E. coli* cells are resuspended in binding buffer (50 mM Tris, pH 7.9, 500 mM NaCl and 10 mM imidazole), lysed with a high-pressure homogenizer and sedimented for 30 min to remove the debris. The supernatant lysates are purified on Ni-NTA agarose beads (QIAGEN). After extensive washing with binding buffer, the proteins are eluted with His$_6$ elution buffer (50 mM Tris, pH 7.9, 500 mM NaCl and 500 mM imidazole), then further purified with a HiPrep 26/60 Sephacryl S-200 HR column (GE Healthcare, 17-1195-01) on an AKTA purifier (GE Healthcare), and finally eluted with a buffer containing 20 mM HEPES, pH 7.5, 500 mM NaCl. The purified proteins are concentrated using centrifugal filters (Millipore).

The phase separation of 6xPSD is induced by simply mixing the six protein components in the buffer solution at the designated concentration of 5 μM each without adding any crowding agent. The phase separation of PGL-3 is induced by diluting the protein solution with an NaCl-free buffer to a final NaCl concentration of 75 mM and a final PGL-3 concentration at 0.8 mg/mL.

### Atomic force microscopy

The mechanical measurements of 6xPSD are conducted using an atomic force microscope (AFM) (MFP-3D, Asylum Research) with a colloidal probe. The colloidal probe is assembled by gluing a glass sphere (of radius $R_1$ ~ 7.5 μm) onto the free end of a tipless cantilever (CSC38, tipless, MikroMasch) with a nominal spring constant ~0.09 N/m. The actual spring constant of the cantilever is calibrated in situ using the power spectrum density method[51]. Prior to each mechanical measurement, the colloidal probe is coated with a thin layer of poly-L-lysine-graft-poly-ethylene-glycol (PLL-g-PEG) (SuSoS AG) to reduce adhesion between the probe surface and PSD droplets and avoid permanent absorption of proteins on the probe surface. The coating is prepared by immersing the AFM probe in the 0.5 mg/mL PLL-g-PEG solution for 3 h after the probe surface is plasma activated using a plasma cleaner (PDC-32G, Harrick Plasma) at a high RF power under the pressure of ~1 mbar for ~10 min.

Before the AFM measurements, the six protein components are diluted with the buffer solution and mixed to the designed combination and concentration (5 μM each) to induce phase separation. Then, the 6xPSD solution is added onto a clean glass coverslip and is transferred into a closed AFM chamber to reduce evaporation during the AFM measurements. All of the mechanical measurements are made within the period between 30 min and 2.5 h after the condensate formation. The first 30 min is used to ensure that most of the suspending condensate droplets settle down to the coverslip surface, avoiding their influence on the AFM laser propagation through the solution. The longest duration of 2.5 h is chosen to avoid possible aging effects of the condensate. The mechanical measurements of 6xPSD remain consistent in the first 3 h after the condensate formation. After the three hours, the aging-induced stiffening[5] begins to affect the mechanical characterization and subsequent time evolution considerably.

In the stress relaxation measurements, the probe is brought into contact with the 6xPSD droplet at $v \simeq 100$ μm/s ((i) approach) and held to maintain a fixed indentation once the measured force reaches the setpoint force $F_0$ ((ii) hold), during which the time evolution of $F(t)$ is recorded (see Fig. 1d). Here, the indentation loading velocity is particularly chosen to be $v \simeq 100$ μm/s, which is high enough to avoid a convoluted relaxation during approach[54], while avoiding undesirable hydrodynamic effects. The hydrodynamic force introduced by moving the colloidal probe could be estimated from $f = 6\pi\eta Rv$ ~ 0.01 nN, with $\eta$ being the viscosity of the solution, which is much smaller than mechanically applied force $F_0$. Meanwhile, the indentation time during

the approach is ~1 ms, which is shorter than the typical resolved exponential decay time $\tau_1 \simeq 3$ ms. This ensures that the perturbation does not affect the short-time exponential decay in the relaxation. The typical setpoint force is chosen to be $F_0 = 1–3$ nN to introduce an indentation $\delta = 0.1–0.2\,\mu$m. As a comparison, the typical height of a 6xPSD droplet is around 2 μm, measured by the difference of AFM contact points on the droplet and substrate. The fixed indentation is maintained at about (10–20)% of the droplet height so that the relaxation measurements fall in the small strain limit of 6xPSD.

In the force-indentation measurements, the probe is first brought into contact with the 6xPSD droplet and continuously compresses the droplet at velocity $v$ ((i) approach). When the resulting force reaches the setpoint force $F_0$, the probe starts to move away from the droplet and retract from the probe surface at the same velocity ((ii) retract). AFM thus records the force and indentation during the compression (see Fig. 1g). The force-indentation measurements are performed with various loading velocities $v = 5, 10, 20, 50\,\mu$m/s and a fixed setpoint force $F_0 = 1–3$ nN.

### Fluorescent imaging and fluorescence recovery after photobleaching (FRAP) assay

The differential interference contrast (DIC) and fluorescent images shown in Fig. 3a are obtained using an upright fluorescence microscope (Nikon Ni-U). The FRAP measurements are performed using a Zeiss LSM 880 confocal microscope. In both experiments, only the protein to be imaged and measured is labeled by Cy3 to avoid possible signal cross-talking between different channels. The 6xPSD solution is imaged in a homemade flow chamber, which is made by a coverslip and a glass slide in parallel with each other. The two glass plates are separated by a spacer made of double-sided tape. After mixing of the six PSD proteins, the solution is allowed to equilibrate at room temperature for about 30 min to avoid drifting during the measurement due to floating droplets. The time required for stabilization of the 6xPSD solution is consistent with the AFM measurements. In the imaging assays, the fluorescence-labeled proteins are further diluted with the corresponding unlabeled proteins in the same buffer to a desired labeling ratio of 1%.

In the FRAP experiments, the labeled protein is bleached using a laser of wavelength 561 nm. The bleaching area is set to cover the whole droplet (see Fig. 3b). The whole-droplet FRAP is chosen for the convenience of comparing the results with the previous measurements[5]. The bleached zone in the fluorescent droplet has a cylindrical shape with radius $r$ in the range of 3–6 μm. The confocal imaging section of the cylinder has a thickness ~0.2 μm, and its vertical position (i.e., the z-position of the imaging plane) is set at the middle plane of the condensate droplets, which is away from the glass substrate to avoid possible substrate effects on the FRAP[5]. For each sample batch, two or three droplets are bleached simultaneously and are imaged with their fluorescence intensity recorded while another droplet not being bleached in the same capture frame is chosen as a reference.

The final fluorescence intensity $I(t)$ of the sample normalized in the range netween 0 and 1 is defined as:

$$I(t) = \frac{\beta(t)I_{\text{net}}(t) - I_{\text{net}}(0^+)}{I_{\text{net}}(0^-) - I_{\text{net}}(0^+)}, \qquad (6)$$

where $I_{\text{net}}(t)$ (=raw intensity – background intensity) is the net fluorescence intensity of the bleached droplet as a function of time $t$, $I_{\text{net}}(0^-)$ is the fluorescence intensity averaged with the last two frames just before the bleaching, and $I_{\text{net}}(0^+)$ is the averaged intensity right after the bleaching. The time $t = 0$ denotes the moment of bleaching. The factor $\beta(t) = I_{\text{ref}}(t)/I_{\text{ref}}(0)$, with the subscript ref referring to the unbleached reference droplet, is introduced to correct the effect of unrelated background fluctuations of the fluorescence intensity

resulting from slight drifts of the imaging plane and other sources. By fitting the measured $I(t)$ to Eq. (5), we obtain the value of the two fitting parameters; one is the diffusion coefficient $D$ and the other is the saturation value $A$ for the labeled protein (see Supplementary Section I.A for details).

### Fluorescence intensity quantification (FIQ) assay and absolute protein concentration determination

To convert the measured fluorescence intensity to the absolute concentration of proteins, FIQ is performed to determine the quantitative relation between the fluorescence intensity and fluorophore concentration, as previously described[5]. Here, we use the Cy3-labeled Shank3 solution with different indicated concentrations (5 μM, 10 μM, and 20 μM) to generate a calibration curve. The exact concentration of the Cy3-Shank3 solution is calibrated beforehand by a spectrophotometer (NanoDrop One, ThermoFisher). The Cy3-labeled Shank3 solution is imaged using a confocal microscope (Zeiss LSM 880) together with the same sample chamber used for the FRAP experiments. The fluorescence intensity of the solution at each protein concentration is measured in the z-stack mode with a 0.2 μm z-interval for a given set of imaging parameters including the laser power, detector gain, imaging size and resolution.

The 6xPSD droplets with one protein component labelled by Cy3 are then imaged in the z-stack mode using the same imaging parameters as those used in the calibration measurement. Only one protein component in the 6xPSD is labelled each time at a low label ratio of 1% to avoid signal crosstalk. The 3D scanning starts at the coverslip slice and ends at the top of the droplet. We select the fluorescence layers entirely within the droplet to calculate the fluorescence intensity. In these layers, the fluorescence intensities remain nearly constant (and they are also the brightest layers), indicating that the proteins in the droplets are homogeneous and the microscopic molecular structures have dimensions smaller than the confocal microscope resolution (typically < 100 nm). By averaging the intensity images of these layers and comparing them with the calibration curve, we obtain the absolute concentration of each protein component (see Supplementary Section I.B for details).

### Phase separation centrifuge assay

The 6xPSD solutions used for centrifuge assay are prepared in a buffer containing 50 mM Tris, 100 mM NaCl, 1 mM EDTA, and 1 mM DTT (pH 8.0). Allowing 10–30 min for equilibrium after mixing, the phase-separated 6xPSD solutions are centrifuged at $16,873 \times g$ and 25 °C for 10 min, after which the supernatant (dilute phase) and pellet (condensed phase) are immediately separated. The pellet is thoroughly re-suspended in the same buffer with a final volume of 50 μL. The six protein components in the final solution of the pellet and supernatant are then analyzed by electrophoresis (SDS-PAGE) with Coomassie blue staining (see Fig. 3f). The band intensities are subsequently used for quantification of the protein concentrations.

Because the NR2B band obtained with Coomassie blue staining appears rather weak and vague, we use the silver staining to improve the visualization and quantification of the NR2B band (see Supplementary Fig. 5 and Supplementary Section I.C for details). In the image analysis desscribed below, Supplementary Fig. 5 is used to quantify the NR2B band intensity and Fig. 3f is used to quantify the band intensity of the other five protein components. The ImageJ software is used to determine the band intensities, $(I_p)_i$ and $(I_s)_i$, respectively from the pellet and supernatant of the $i$th protein component. The pellet percentage $p_i$ of the $i$th protein component is then determined as

$$p_i = \frac{(I_p)_i}{(I_p)_i + (I_s)_i}. \qquad (7)$$

**Table 3 | Obtained protein concentrations in the condensed phase of 6xPSD using FIQ and centrifuge assay**

| Protein components | PSD-95 | GKAP | Shank3 | Homer3 | NR2B | SynGAP |
|---|---|---|---|---|---|---|
| Concentration by FIQ (mM) | 0.85 | 1.74 | 1.58 | 0.95 | 0.90 | 1.23 |
| Concentration by centrifugation (mM) | 1.02 | 1.48 | 1.58 | 1.85 | 1.72 | 1.05 |
| Mean (mM) | 0.93 | 1.61 | 1.58 | 1.40 | 1.31 | 1.14 |
| Uncertainties (mM) | 0.09 | 0.22 | 0.09 | 0.45 | 0.41 | 0.09 |

Fluorescence intensity quantification (FIQ) is used to determine the concentration of each protein component individually. The centrifuge assay is used to determine the relative ratio of protein concentrations among the six components. Equation (9) is used to convert the obtained relative ratio of protein concentrations to the absolute concentration of the five protein components (PSD-95, GKAP, Homer3, NR2B, and SynGAP) using the molar concentration $n_{Shank3}$ = 1.58 (mM) of Shank3 obtained by FIQ as a reference.

The molar concentration of the $i$th protein component in the condensed phase (pellet) is defined as

$$(n_p)_i = \frac{(N_p)_i}{V_p} = \frac{N_{input}p_i}{V_p}, \qquad (8)$$

where $(N_p)_i$ is the molar amount of the $i$th protein component in the pellet, $V_p$ is the pellet volume before re-suspension, and $N_{input}$ is the input molar amount of each protein component, which is known upon mixing. Because the pellet volume $V_p$ is too small to be measured accurately, Eq. (8) cannot be used directly to determine the absolute concentration of each protein component. As a result, the centrifuge assay can only be used to determine the relative ratio of protein concentrations among the six components of the 6xPSD condensate.

Given that $V_p$ and $N_{input}$ are the same for all the protein components, we obtain the following relation from Eq. (8)

$$(n_p)_i = n_{ref} \frac{p_i}{p_{ref}}, \qquad (9)$$

where the subscript ref refers to a reference component and $i$ runs for other five protein components, respectively. Equation (9) can be used to convert the measured relative ratio of protein concentrations to the absolute concentration of each protein component when the absolute concentration of the reference component is known, say, by FIQ. Because the fluorescence property of Shank3 has been quantified extensively in the previous confocal experiments[5], we choose its molar concentration $n_{Shank3}$ as the reference.

In Table 3, we list the molar concentrations of all six protein components in the condensed phase obtained by FIQ and centrifuge assay. The concentration values obtained by the two methods are in general agreement with the maximal uncertainties no more than 32%. In Table 2, we use the mean value of the protein concentrations obtained by the two methods to avoid possible systematic errors from each method. The measurement uncertainties given in Table 3 are taken from either the standard deviation of the FIQ measurements (for PSD-95, GKAP, and Shank3) or the difference between the two methods (for Homer3, NR2B, and SynGAP), whichever is greater.

## Droplet coalescence measurements

To obtain the interfacial properties of the condensate droplets in water, we conduct the droplet coalescence measurements using the same AFM setup together with bright field imaging. The AFM tip moves around to perturb the fluid and put two 6xPSD droplets in contact. Under the action of interfacial tension $\gamma$, the two condensate droplets in contact gradually merge into a single spherical droplet to minimize the surface energy, as shown in Fig. 2c. A good measure of the surface area reduction is the droplet aspect ratio AR during the coalescence. Figure 2d shows the obtained AR($t$) as a function of coalescence time $t$.

The obtained value of $\gamma$ for the 6xPSD droplets falls in the range of $\gamma$, $10^{-7}$–$10^{-4}$ N/m, reported previously for various biomolecular condensates[56,58,87–89]. An estimated value of $\gamma \lesssim 3 \times 10^{-6}$ N/m was also reported for the PGL-3 condensate[20,56]. For the interfacial tension of the 6xPSD droplets in water, $\gamma \simeq 0.2 \times 10^{-6}$ N/m, the largest possible capillary force (i.e., the contact line force) acting on the AFM probe is given by $\pi d \gamma \simeq 9.4$ pN, where we have used the AFM probe diameter $d = 15$ μm to estimate the contact line length. The actual contact line length is much smaller than $\pi d$, because the indentation $\delta$ involved is only a fraction of a micrometer (see Fig. 1h). As shown in Fig. 1h, the measured force-indentation $F(\delta)$ (and force relaxation $F(t)$) are all in the 1–3 nN range. Because the interfacial force is much smaller than the obtained $F(t)$ and $F(\delta)$, it becomes evident that contributions to the obtained $F(t)$ and $F(\delta)$ must result from the bulk of the 6xPSD droplets and the interfacial effect on the obtained $F(t)$ and $F(\delta)$ is negligibly minor.

## Reporting summary

Further information on research design is available in the Nature Portfolio Reporting Summary linked to this article.

## Data availability

Source data that support the findings of this study (such as figure source data) are provided with this paper in the Source Data file. Unless otherwise stated, all the data supporting the results of this study can be found in the article, supplementary, and source data files. The raw data are available from the corresponding authors upon request. Source data are provided with this paper.

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

## Acknowledgements

We thank Profs. Hong Zhang and Zheng Wang at the Institute of Biophysics, Chinese Academy of Sciences for providing the PGL-3 samples. This work was supported in part by the Key Research Program of Chinese Academy of Sciences under grant no. ZDBS-ZRKJZ-TLC002 (D.G.), NSFC under grant no. 12372267 (D.G.), and the Strategic Priority Research Program of CAS under grant no. XDB0620102 (D.G.), by the Hong Kong RGC under grant nos. 16300920 (P.T.), 16300421 (P.T.), 16104518 (M.Z.), and 16101419 (M.Z.), by the National Natural Science Foundation of China under grant no. 82188101 (M.Z.), and by the Minister of Science and Technology of China under grant no. 2019YFA0508402 (M.Z.).

## Author contributions

D.G., M.Z. and P.T. designed research; Z.L. and B.J. performed research; Z.L. conducted AFM experiments and AFM data analysis; B.J. and Z.L. conducted fluorescent and centrifuge experiments; D.G. and X.C. performed early measurements; all authors analyzed and interpreted data; Z.L., D.G., and P.T. drafted the paper with inputs from B.J., X.C. and M.Z.; M.Z. and P.T. coordinated the project.

## Competing interests

The authors declare no competing interests.
