## [Transparent Peer Review file · Nature Communications]

Emergent mechanics of a networked multivalent protein condensate

Corresponding Author: Professor Penger Tong

Version 0:

Reviewer comments:

Reviewer #1

(Remarks to the Author)

In the manuscript "Emergent mechanics of a networked multivalent protein condensate", Liao and colleagues combine AFM and fluorescence microscopy tools to quantify the material properties of the 6xPSD condensate system. They hypothesize that multi-component condensates should exhibit material properties that are more complex than single component condensates that can be described as Maxwell viscoelastic fluids characterized by a single relaxation time. Data is presented that indeed supports multiple relaxation times of 6xPSD condensates by AFM (Fig. 1). They go on to further characterize the network using FRAP and quantitative imaging (Fig. 2). The authors then further discuss their findings and conclude that these condensates fit the model of cross-link-governed dynamics (CGD), which take into account the relative binding affinities between components previously measured (Fig. 3).

This is overall a very exciting research area, and approach. However, there are several very important issues that would need to be addressed before further considered for publication.

Major Issues:

1. The authors dutifully reference the prior works (Jawerth et al and Al Shareedeah et al) that used optical tweezer methods on simpler systems to support a Maxwell model as an introduction to the novelty of their work here. Since the main argument for the novelty of this work relies on these multiple components arising from the multi-component system – as is stated in the title – it is imperative that the authors apply their methods to one of these simpler systems – to show that indeed these timescales emerge from the multi-component system, and NOT from the methodology or the measurement. This is crucial.
2. Regarding the quantification of protein concentration that is relied on for many aspects of the theory – and thus very important:
 - a. Fluorescence method: The authors mention that they use 1% labeled protein in the condensate. This allows them presumably to stay within the linear range of intensity...so for example if they quantify ~10uM labeled protein within the condensate, they then convert this to ~1mM protein. for this to work several things would need to be assumed: 1) they know what their original labeling efficiency is – which was not mentioned and should be calculated and discussed in the methods, and 2) that the fluorescent label doesn't influence the partitioning of the protein. This point could perhaps be tested by adjusting the % labeled protein included and seeing if you backout the same values. Finally, 3) how do the authors take into account the change in refractive index within the condensate and/or whether the crowded environment within the condensate quenches – (or even enhances!) – the fluorescence? There are several important controls that should be done.
 - b. Centrifugation method: It is not clear to this reviewer how you are able to backout the concentration in the pellet without having an accurate measurement of the original pellet volume -- which is very small, and very difficult to do!
 - c. Finally, it is mentioned that the concentration values displayed in Table II are averaged/combined from these two methods. It is important to include the raw numbers for these in the supplementary so that they can be compared...and to see how well they are in agreement or not.
3. Regarding the FRAP
 - a. Why did the authors choose to FRAP the whole condensate? The authors cite the work by Taylor et al, which I believe

demonstrates that whole FRAP, i.e. when the ROI size is similar to the droplet size, leads to significant errors in diffusion coefficient. Can you please expand on the FRAP modeling? Also, the droplets are quite big, so it should be rather simple to do a smaller bleach spot. Additionally, it is also known that whole droplet FRAP is likely combining diffusion rates within the droplet as well as exchange in/out of droplet – which is something that I don't believe is accounted for in the discussion.

b. The repeated FRAP cycling is actually very cool

4. Regarding Figure 3 and the discussion of the CGD model: This is an exciting proposal and discussion – however, it should be strengthened by testing this prediction by specifically altering the K_d of some of the components and measuring the mechanical outputs.

Additional questions/comments:

1. Are all 6 proteins required for phase separation in the PSD system? Or are just the scaffold proteins necessary? It would be great if you could introduce the 6xPSD a little bit more in this paper – specifically, which proteins are required – and summarize the main points from prior work that the reader should be aware of.
2. In the 2nd paragraph in the introduction, when citing works that using particle tracking, at least two important works should be cited in addition to the current ref's 15-19, specifically Zhang et al 2015 DOI: 10.1016/j.molcel.2015.09.017 and Elbaum-Garfinkle et al 2015 10.1073/pnas.1504822112
3. Is this really a “newly developed technology” ?
4. I don't see a description for how droplets are made/triggered to form? Are all the components simply mixed?
5. In the discussion of mesh size estimates, should also cite Wei et al 2017 DOI: 10.1038/nchem.2803
6. What is the reference for the mesh size calculation used?
7. Regarding discussion of the exponential relaxation in relation to poroelastic relaxation...can you elaborate on why it should take longer for solvent to redistribute, than for local mobile protein concentrations around the network? That doesn't seem intuitive to this reviewer!

Reviewer #2

(Remarks to the Author)

The authors characterize the viscoelastic properties of a protein condensate formed by six different postsynaptic density protein (called 6xPSD), by means of atomic force microscopy and fluorescence microscopy. They first measure the viscoelastic properties of the multi-protein condensate measuring the stress relaxation modulus. To further investigate the underlying protein network, the authors use fluorescence labelling to track the contribution of each component to the viscoelastic properties of the system. Interestingly, they are able to quantify the proportion of mobile vs. immobile proteins within the condensate.

The work is concise and well-written, the characterization of the viscoelasticity is quite complete as they provide both general observations and local analysis of each individual component. However, the work may be not applicable to more homogenous liquid-like systems and thus it could deserve publication in Nature Communications after some major changes.

Major issues:

1- The main problem with this work is the lack of applicability to other systems with less components. The work seems designed to this specific system made of many proteins that form an underlying network. First, I would suggest the authors to justify why they use the system of 6xPSD, some reasons should be provided in the introduction. Also, the other system they characterize seems rather like the original provided in the main text. Second, I wonder to what extent this method for characterizing viscoelasticity is applicable to more simple liquid-like protein condensates such as those studied in other works as these:

Jawerth, L., Fischer-Friedrich, E., Saha, S., Wang, J., Franzmann, T., Zhang, X., ... & Jülicher, F. (2020). Protein condensates as aging Maxwell fluids. *Science*, 370(6522), 1317-1323., Fisher, R. S., & Elbaum-Garfinkle, S. (2020). Boeynaems, S., Holehouse, A. S., Weinhardt, V., Kovacs, D., Van Lindt, J., Larabell, C., ... & Gitler, A. D. (2019). Spontaneous driving forces give rise to protein– RNA condensates with coexisting phases and complex material properties. *Proceedings of the National Academy of Sciences*, 116(16), 7889-7898.

Alshareedah, I., Borchers, W. M., Cohen, S. R., Singh, A., Posey, A. E., Farag, M., ... & Banerjee, P. R. (2023). Sequence-specific interactions determine viscoelasticity and aging dynamics of protein condensates. *bioRxiv*, 2023-04.

Tunable multiphase dynamics of arginine and lysine liquid condensates. *Nature communications*, 11(1), 4628.

The works presented above use different techniques. To what extent the AFM and FRAP can provide compatible results for the systems studied in this work? The system reminds more to a melt of associative polymers rather than a protein condensate due to the underlying protein mesh. You can check the classic works of Rubinstein and Semenov such as Semenov, A. N., & Rubinstein, M. (1998). Thermoreversible gelation in solutions of associative polymers. 1. Statics. *Macromolecules*, 31(4), 1373-1385.

2- The introduction to the problem is poor, the authors should introduce the problem and the importance of characterizing the

viscoelasticity and why to study the system of 6xPSD. The work lack theoretical approaches to the problem or simulation works that have recently contributed to the field. Here there are a few works they can cite:

Devarajan, D. S., Wang, J., Szala-Mendyk, B., Rekhi, S., Nikoubashman, A., Kim, Y. C., & Mittal, J. (2024). Sequence-dependent material properties of biomolecular condensates and their relation to dilute phase conformations. *Nature Communications*, 15.

Tejedor, A. R., Collepardo-Guevara, R., Ramírez, J., & Espinosa, J. R. (2023). Time-dependent material properties of aging biomolecular condensates from different viscoelasticity measurements in molecular dynamics simulations. *The Journal of Physical Chemistry B*, 127(20), 4441-4459.

Tejedor, A. R., Garaizar, A., Ramírez, J., & Espinosa, J. R. (2021). 'RNA modulation of transport properties and stability in phase-separated condensates. *Biophysical Journal*, 120(23), 5169-5186.

3- The authors provide a schematic depiction of the protein network structure in Fig. A-B. It seems to be unclear how they infer that specific structure with the results they provide in the current work. Is this something well-known? If so, please provide the reference. Apart from that figure 3 is quite informative despite it does not include any new analysis. In addition, the authors have performed the same analysis to a second sample (S2), bringing these results (not necessarily all of them) to the main text could improve the impact of the work.

4- The work does not estimate the viscosity of the condensate. In the introduction the authors include a discussion about the complex modulus and the crossover of G' and G'' in the low frequency regime when the condensate behaves as a gel.

Could the authors estimate the elastic and viscous modulus of the studied system? See for instance:

Alshareedah, I., Borcherds, W. M., Cohen, S. R., Singh, A., Posey, A. E., Farag, M., ... & Banerjee, P. R. (2023). Sequence-specific interactions determine viscoelasticity and aging dynamics of protein condensates. *bioRxiv*, 2023-04.

Additionally, the authors can cite the book of Rubinstein (*Polymer Physics*) as a reference for theoretical expressions.

5- In Fig. 1C the exponential and potential decay are confusing. Is that a fit to the data? The sum of both contributions clearly does not fit the experimental data. Authors should be clearer on that. The discussion of this figure is confusing. Maybe split the figure in two could help to guide the reader. Just out of curiosity, could you calculate the hysteresis from Fig. 1C as a function of the loading speed? This could be a complementary analysis.

Minor comments:

- In some cases, the notation is a bit confusing. Again, I don't understand why they introduce the complex modulus if they don't perform any analysis in that respect. Could they note the relaxation modulus as $G(t)$ instead of $E(t)$?
- In Eq. (3) it would help the reader to provide the explicit expression for the correction factor $C(t)$.
- The authors state that the fit of Eq. (4) lead to a universal master curve (Fig. S2) when plotting the $I(t)/A$ vs. t/τ . This is trivial as long as the data can be fitted to Eq. 4, so it is more valuable to discuss why the data follow Eq. (4).

Reviewer #3

(Remarks to the Author)

Reviewer #4

(Remarks to the Author)

This manuscript discusses probes the mechanical response of a six-component biomolecular condensate using AFM and FRAP.

AFM relaxation measurements reveal a short term (<few ms) exponential relaxation followed by a power-law decay over a few decades. Force-indentation measurements show a rate dependent indentation with a strong adhesive force on pull-off. FRAP experiments reveal a significant immobile fraction inside the droplets. The authors conclude that the mechanical response is viscoelastic, with a short-time Maxwell-like response due to the diffusion of the mobile species, and a long-time power-law response reflecting slow network re-arrangements.

While the data appears to be of high quality, I am skeptical of its physical interpretation.

- AFM a) AFM data has contributions from interfacial forces b) the compressional deformation of AFM mixes poroelastic and viscoelastic responses, c) measured forces are very sensitive to the contact geometry, which is unobserved and d) the data is acquired in a limit where the contact radius is comparable or even bigger than the sample thickness, where results are hard to interpret and very sensitive to this ratio. It seems that particle tracking microrheology would be a much more appropriate choice for probing viscoelasticity.

- I think the FRAP data is tricky to interpret because a) the sample is very thin and interactions of the protein with the probe or surface of the sample holder could immobilize protein.

I am skeptical of the hypothesized relevance to cells, as described at the end of the conclusion. The main issue is that the system has six components (plus buffer). Physical properties will vary dramatically across this high-dimensional phase diagram. There is no description in this manuscript of how the composition of the protein droplets in these studies compares to those in cells.

Further concerns on the physical interpretation of the data

1. I am concerned that the AFM method is sensitive not only to bulk, but also surface properties of the droplets.
 - a. The determination of the time dependent modulus assumes that the contact area is constant. But contact lines are infamously slow to relax. There is no reason, a priori, to assume that the dynamics of relaxation are due to a bulk relaxation, instead of contact line motion. Do the authors have data on the contact area over time?
 - b. The authors observe adhesive forces comparable to the total force on indentation. I appreciate the control experiments with different surface coatings, but I am not convinced by it. Given the huge adhesion, it's clear that the droplet shape and contact area must be strongly affected by interfacial forces in ways that are not accounted for by the Hertz model
2. Poroelasticity is discounted much too quickly. Poroelastic time constants are sensitive to geometry, network properties and solvent properties. So, randomly picking some timescales from poroelasticity papers is insufficient to rule out this mechanism. The best way to rule it out would be to look at relaxation timescale versus indentation depth or radius of indenter.
3. The authors jump too quickly to viscoelasticity. The null hypothesis of liquid-liquid phase separation (which they invoke in the first sentence of the abstract) is that the fluids are Newtonian. The manuscript would be much more compelling if their analysis started by showing that the observed behavior is inconsistent with Newtonian rheology. Given the strength of the interfacial forces (as seen in the adhesion), and likely contact line pinning and relaxation, I think this is a hard case to make. The droplet shape should have an exponential relaxation determined by the competition of its size and the capillary velocity. Contact lines can relax very slowly. Further, because the indenter is so close to the surface, and the sample is so thin, one would have to also rule out slow drainage of the lubricating film. In the absence of measurements of the contact radius versus indentation, it will be hard to rule out Newtonian fluids for the force-indentation curve.

At the end of the day, while the experiments are nicely done. I think there are too many ambiguities in the interpretation to give a definitive result. If this was the only method to try to measure these things, I might be more lenient. Since other methods exist to more reliably and easily measure rheology (particle tracking microrheology), I am very reluctant to put much emphasis, as a community, on these results. Do the authors have some microrheology data on this system to compare to? Is there any information in the AFM test that is not available from microrheology?

Version 1:

Reviewer comments:

Reviewer #1

(Remarks to the Author)

The revisions look great and I recommend its publication.

Reviewer #2

(Remarks to the Author)

The revised version of the manuscript is now ready for publication.

Reviewer #5

(Remarks to the Author)

I have been asked by the editor to evaluate the response of authors to the comments on AFM micro-rheology by reviewer 4.

To start with, I fully agreed with reviewer 4 on the many detailed points where the analysis of the AFM micro-rheology may go wrong, and which were not yet adequately discussed in the original ms. At the same time, I am impressed by the rebuttal of the authors. They have gone to great lengths to do the additional experimental checks suggested by the reviewer, and rebutted the various comments using in my opinion convincing arguments and estimates.

In short, in my opinion the rebuttal of the authors to the legitimate points on AFM micro-rheology by reviewer 4 is adequate.

Response to Referee #1:

The authors wish to thank the referee for his/her careful review and constructive suggestions for this work. These suggestions have helped us considerably improve and strengthen the manuscript. The following are our responses to each of the referee's comments (marked in blue). All the changes made in the main text and supplementary materials are marked in red.

Reviewer #1 (Remarks to the Author):

In the manuscript “Emergent mechanics of a networked multivalent protein condensate”, Liao and colleagues combine AFM and fluorescence microscopy tools to quantify the material properties of the 6xPSD condensate system. They hypothesize that multi-component condensates should exhibit material properties that are more complex than single component condensates that can be described as Maxwell viscoelastic fluids characterized by a single relaxation time. Data is presented that indeed supports multiple relaxation times of 6xPSD condensates by AFM (Fig. 1). They go on to further characterize the network using FRAP and quantitative imaging (Fig. 2). The authors then further discuss their findings and conclude that these condensates fit the model of cross-link-governed dynamics (CGD), which take into account the relative binding affinities between components previously measured (Fig. 3).

This is overall a very exciting research area, and approach. However, there are several very important issues that would need to be addressed before further considered for publication.

Major Issues:

1. The authors dutifully reference the prior works (Jawerth et al and Al Shareedah et al) that used optical tweezer methods on simpler systems to support a Maxwell model as an introduction to the novelty of their work here. Since the main argument for the novelty of this work relies on these multiple components arising from the multi-component system – as is stated in the title – it is imperative that the authors apply their methods to one of these simpler systems – to show that indeed these timescales emerge from the multi-component system, and NOT from the methodology or the measurement. This is crucial.

Following the referee's suggestion, we repeated the AFM measurements for the one-component PGL-3 condensate, which is the same as that used by Jawerth *et al.* (Science, 2020) and was prepared similarly. Figure R1 below compares the obtained force relaxation curves between PGL-3 (red circles) and 6xPSD (black squares). It is seen that the measured force relaxation $F(t)/F_0$ (proportional to the modulus $E(t)$) for the PGL-3 droplets (red dots) decays to zero much faster than that for 6xPSD and can be well described by a simple exponential decay without power-law relaxation. The single exponential decay in the relaxation modulus $E(t)$ is a hallmark of Maxwell fluids (see, e.g., p.168 in Ref. 28 and p.284 in Ref. 29). The results shown in Fig. R1 thus confirm that the PGL-3 condensate is of Maxwell type, as previously reported. They also demonstrate that our AFM methodology of measuring the relaxation modulus $E(t)$ works, and the power-law relaxation in the measured $F(t)/F_0$ is an emergent property of 6xPSD.

In the revised manuscript, we added the PGL-3 data in Fig. 1(F) and a discussion paragraph in the main text (p.4, left column, 2nd paragraph). Details about the sample preparation were added to the Supplementary Information (SI) (p.1, Sec. I.A, right column, 1st paragraph).

Fig. R1. Comparison of the normalized force relaxation $F(t)/F_0$ between the PGL-3 droplets (red circles) and 6xPSD droplets (black squares). The solid lines show the fits of Eq. (1) in the main text to the data points. The fit to the PGL-3 data gives $C_1 = 1$, $\tau_1 = 2.8$ ms, and $C_2 = 0$.

We also converted the measured relaxation modulus $E(t)$ to the complex shear modulus $G^*(\omega)$, previously used to describe the viscoelastic properties of PGL-3. The obtained $G^*(\omega)$ for PGL-3 is found to be in good agreement with the previous results by Jawerth *et al.* (Science, 2020) and is well described by the Maxwell model in the frequency domain (see SI Fig. S8(A)).

In the revised manuscript, we added a discussion on the relation between the relaxation modulus $E(t)$ in the time domain and $G^*(\omega)$ in the frequency domain (p.4, left column, 1st paragraph). A new figure and a new discussion subsection were added in SI (p.8-10, Fig. S8 and Sec. III.A).

2. Regarding the quantification of protein concentration that is relied on for many aspects of the theory – and thus very important:

a. Fluorescence method: The authors mention that they use 1% labeled protein in the condensate. This allows them presumably to stay within the linear range of intensity...so for example if they quantify $\sim 10\mu\text{M}$ labeled protein within the condensate, they then convert this to $\sim 1\text{mM}$ protein. for this to work several things would need to be assumed: 1) they know what their original labeling efficiency is – which was not mentioned and should be calculated and discussed in the methods, and 2) that the fluorescent label doesn't influence the partitioning of the protein. This point could perhaps be tested by adjusting the % labeled protein included and seeing if you backout the same values. Finally, 3) how do the authors take into account the change in refractive index within the condensate and/or whether the crowded environment within the condensate quenches – (or even enhances!) – the fluorescence? There are several important controls that should be done.

The fluorescence intensity quantification (FIQ) method used in this experiment to determine the absolute concentration of the protein condensate has been fully characterized in a previous Cell paper (Zeng *et al.*, Cell **174**(5), 1172-1187 (2018)). It was also used in two recent studies: Wu *et al.* Molecular Cell, **73**(5), 971-984 (2019) and Lin *et al.* Biophysical Journal, **121**(1), 157-171 (2022). The following are our answers to the referee's specific questions.

1) Before the confocal imaging, we calibrate the total and labeled protein concentrations using the Nanodrop One spectrophotometer (Thermo Fisher). We then mix the labeled protein with the unlabeled one to achieve the desired labeling efficiency of 1% for all the fluorescence image measurements. This procedure was added in Methods as suggested (SI p.1, Sec. I.A, left column, last paragraph).

- 2) From our extensive working experience with the 6xPSD samples, we found no evidence that the fluorescent dye influences the partition of the 6xPSD condensates. To further quantify this answer, we labeled PSD-95 with three different ratios of 0.5%, 1%, and 2%, respectively, for fluorescence imaging of the 6xPSD condensate (see Fig. R2(A) below). The obtained fluorescent intensity fits well to a linear function of the labeling ratio (solid line in Fig. R2(B) with a zero intercept), and the obtained intensity ratios are close to the expected ones of 1:2:4. Following the Referee's suggestion, we added Fig. S4 and a discussion paragraph in SI (p.3, Sec. I.C, right column, last paragraph).

Fig. R2. Fluorescence images and the resulting intensity of the 6xPSD droplets with different label ratios. (A) Confocal images of the 6xPSD droplet with 0.5%, 1%, 2% PSD-95 labeled by Cy3. (B) Measured fluorescence intensity as a function of the labeling ratio. The data points are obtained, respectively, from 22 (0.5%), 28 (1%), and 22 (2%) droplets. The error bars show the standard deviation of the measurements. The solid line shows a linear fit to the data points with a zero intercept.

- 3) If the fluorescence changes with the variation in the refractive index of the condensate, the measured fluorescence would depend on the sample thickness (or droplet height). In a previous study of the 6xPSD condensate (see Fig. S4B extracted from Zeng *et al.* Cell, **174**(5), 1172 (2018); also shown below), it has been shown that the measured fluorescence does not change with the droplet size. Figure R3 below shows another example from our experiment. Given that the 6xPSD droplets of different diameters have the same contact angle with the substrate, the droplet height changes linearly with the droplet size. We, therefore, conclude that the measured fluorescence intensity does not change with the droplet height (or sample thickness). Based on these measurements, we conclude that the change in the refractive index of the 6xPSD condensate, if any, would not have a measurable effect on the fluorescence. We added a sentence in the revised main text and cited two new references to clarify this point (p.6, right column, 3rd paragraph).

[Figure Redacted]

Fig. S4B from Zeng *et al.* Cell, **174**(5), 1172 (2018).

Fluorescent images (left) and the measured fluorescence intensity at two yellow lines (right). It is shown that the measured fluorescence intensity is the same for large droplets and small droplets, and the fluorescence is uniformly distributed in the droplets.

Fig. R3. Confocal image of two 6xPSD droplets with different sizes (left panel) and the corresponding fluorescence intensity distribution along the yellow line (right panel).

b. Centrifugation method: It is not clear to this reviewer how you are able to backout the concentration in the pellet without having an accurate measurement of the original pellet volume -- which is very small, and very difficult to do!

The referee is quite correct in pointing out that it is difficult to measure the absolute value of the protein concentration using the centrifugation method because the pellet volume is too small to be measured accurately. Here, we only use it to calibrate the relative ratio of protein concentrations among the six components of the 6xPSD condensate. This is possible because the six protein components are contained in the same pellet volume after the centrifugation. Using the absolute value of the Shank3 concentration obtained from the fluorescence as a reference, we convert the relative ratio to the absolute concentration for the other five protein components (PSD-95, GKAP, Homer3, NR2B, and SynGAP).

To clarify this point further, we revised the Table II caption and the discussion in the main text (p.6, right column, last paragraph). In addition, we revised the entire Sec. I.D in SI (p.3-5).

c. Finally, it is mentioned that the concentration values displayed in Table II are averaged/combined from these two methods. It is important to include the raw numbers for these in the supplementary so that they can be compared...and to see how well they are in agreement or not.

Following the referee's suggestion, we added Table S1 in SI (p.5), which lists the raw protein concentration values obtained by the two methods. The concentration values obtained by the two methods generally agree, with maximal uncertainties of no more than 32%. In Table II of the main text, we use the mean value of the protein concentrations obtained by the two methods to avoid possible systematic errors from each method (Other numbers in Table II are also updated accordingly). The measurement uncertainties given in Table S1 (and Table II in the main text) are taken from either the standard deviation of the FIQ measurements (for PSD-95, GKAP, and Shank3) or the difference between the two methods (for Homer3, NR2B, and SynGAP), whichever is greater.

To clarify these points, we added a discussion paragraph in Sec. I.D of SI (p.4, left column, last paragraph).

3. Regarding the FRAP

a. Why did the authors choose to FRAP the whole condensate? The authors cite the work by Taylor et al, which I believe demonstrates that whole FRAP, i.e. when the ROI size is similar

to the droplet size, leads to significant errors in diffusion coefficient. Can you please expand on the FRAP modeling? Also, the droplets are quite big, so it should be rather simple to do a smaller bleach spot. Additionally, it is also known that whole droplet FRAP is likely combining diffusion rates within the droplet as well as exchange in/out of droplet – which is something that I don't believe is accounted for in the discussion.

The referee is correct in pointing out that the whole-droplet FRAP may lead to errors in the diffusion coefficient due to the exchange of the labeled protein molecules in/out of the droplet. The reason we chose the whole-droplet FRAP in this experiment is for the convenience of comparing the results with the previous measurements (Zeng et al., (2018), Cell, 174(5), 1172). A larger bleaching area will also increase the signal-to-noise ratio of the FRAP measurements.

To examine the effect of the bleaching area on the measured recovery time τ , we conducted simultaneous FRAP measurements on two 6xPSD droplets of different sizes with approximately the same bleaching area. One droplet is significantly larger than the bleaching area, and the other is the same as the bleaching area. The final results are shown in Fig. R4 below. The fitted recovery time τ for the whole-droplet FRAP (red curve) is about 32% smaller than when the bleaching area only covers a portion of the droplet (after a minor correction due to a slight difference in the radius r of the bleaching area). One of the experimental errors contributing to the deviation between the two FRAP measurements is that the exchange of the labeled protein molecules in/out of the droplet may cause the whole-droplet FRAP to have a slightly faster recovery time τ . Therefore, the choice of bleaching area in the FRAP measurement becomes a balancing act between increasing the signal-to-noise ratio and reducing the droplet boundary effects. While our FRAP measurements have a (20-40)% standard deviation in the measured diffusion coefficient D , as shown in Table II of the main text, the obtained mean values of D for each protein component in the condensed phase are adequate for supporting the main conclusions of the work.

Following the referee's suggestion, we added Fig. R4 (Fig. S3) and a discussion paragraph in SI (p.3, left column, last paragraph). We also revised the discussion paragraph in SI (p.2, left column, 2nd paragraph). In Table II of the main text, we added the standard deviation in the measured diffusion coefficient D . We also added a reminder note for the effect of the bleaching area on the measured values of D in the revised manuscript (p.6. left column, last paragraph).

Fig. R4. Comparison of the FRAP measurements conducted when the bleaching area covers a whole droplet and only a portion. (A) Time-lapse images of FRAP obtained from 6xPSD droplets with 1% PSD-95 labeled by Cy3. The blue triangular arrows point to a circular bleaching area of radius $r = 5.1 \mu\text{m}$ in a larger 6xPSD droplet. The red triangular arrows point to a circular bleaching area of radius $r = 5.0 \mu\text{m}$, covering a whole droplet. Scale bar: $5 \mu\text{m}$.

(B) Normalized fluorescence intensity $I(t)$ that are obtained from the two bleaching areas shown in (A). The color code used is the same as that in (A). The solid lines show the fits of Eq. (5) in the main text (or Eq. (S2) in SI) to the data points with $A = 0.93$ and $\tau = 619$ s for the blue curve and $A = 0.86$ and $\tau = 451$ s for the red curve.

b. The repeated FRAP cycling is actually very cool.

4. Regarding Figure 3 and the discussion of the CGD model: This is an exciting proposal and discussion – however, it should be strengthened by testing this prediction by specifically altering the K_d of some of the components and measuring the mechanical outputs.

Following the referee’s suggestion, we made several attempts to alter the binding affinity (or the dissociation constant K_d) between the two scaffold proteins, PSD-95 and GKAP. The binding between PSD-95 and GKAP is driven primarily by the specific interactions between the guanylate kinase (GK) domain in PSD-95 and the GK-binding repeats (GBR) domain in GKAP. With the modifications that target the GBR domain in GKAP, such as phosphorylation and amino acid sequence substitution, one may regulate the interaction strength of the PSD-95/GKAP binding and observe its effects on the condensate mechanics.

First, we tried to phosphorylate the scaffold protein GKAP. Upon phosphorylation of a serine residue in the GBR domain, the binding between the phosphorylated GKAP (pi-GKAP) and PSD-95 is enhanced by about 900-fold compared to the un-phosphorylated one (K_d is changed from 176 μM to 0.2 μM ; see Ref. 48 in SI). With pi-GKAP, the 6xPSD droplets became very sticky to the AFM probe, so the measured relaxation force $F(t)$ may contain a considerable contribution from the adhesion at the contact between the upper surface of the protein droplet and AFM probe. As shown in Fig. R5(A) below, the measured force relaxation $F(t)/F_0$ (proportional to the time-dependent modulus $E(t)$) for the 6xPSD with pi-GKAP (red circles) changes considerably compared with the control (un-phosphorylated 6xPSD, black squares). The red circles decay slower than the black squares and level off at large times t . This result suggests that a solid-like permanent network is developed in the condensate so that its modulus $E(t)$ has an asymptotic (constant) value of E_∞ at large t . We speculate that the phosphorylation of GKAP changes the protein-protein interactions so drastically that the network mechanics change significantly. Thus, the network behavior of the 6xPSD with pi-GKAP is more solid-like than the un-phosphorylated one.

Second, we utilized a designed GKAP with its GBR domains substituted by a special amino acid sequence from a peptide DLS that mimics the GKAP phosphorylation (GKAP-DLS). The sequence substitution in GKAP is similar to the phosphorylation of GKAP but affects the PSD-95/GKAP binding in a weaker manner. The binding strength between GKAP-DLS and PSD-95 is increased approximately by 100-fold (K_d is changed from 176 μM to 1.7 μM ; see Ref. 48 in SI). As shown in Fig. R5(B) below, the measured force relaxation $F(t)/F_0$ for the 6xPSD with one DLS substitution (6xPSD-1xDLS, red circles) follows the same two-mode relaxation as the control (black squares). Its power-law relaxation has a slightly smaller exponent α (changed from 0.54 to 0.5) and twice larger relaxation time τ_2 (changed from 2.9 ms to 5.8 ms). When the number of DLS substitutions is increased to three, the binding avidity between GKAP-3xDLS and PSD-95 is further enhanced. The measured $F(t)/F_0$ (6xPSD-3xDLS, blue triangles) decays more slowly and eventually levels off at large times t , similar to the red circles in Fig. R5(A). When the weakest binding of PSD-95/GKAP is enhanced sufficiently, other protein

interactions may start to play a role in determining the elastic response of the 6xPSD at long times.

The above results provide qualitative support to the proposed mechanism that the 6xPSD's power-law response depends sensitively on the PSD-95/GKAP binding. When the binding strength between GKAP and PSD-95 is enhanced by phosphorylation or sequence substitution, the resulting condensates become more solid-like (e.g., the appearance of a permanent modulus component E_∞ at large times). To test the CGD model more quantitatively, one needs to introduce well-controlled and small perturbations to the PSD-95/GKAP binding at a specific targeted site. The changes induced by phosphorylation and sequence substitution appear so drastic that the intrinsic nature of the protein network is changed from a transient to a more permanent network. As the 6xPSD has multiple protein components and numerous targeted sites for specific protein-protein interactions, finding a precise way to fine-tune the interaction strength at a specific targeted site requires a systematic effort to explore an ample parameter space. While this task is important in its own right, it is beyond the scope of this work.

Following the referee's suggestion, we added Fig. R5 (Fig. S18) and a discussion section in SI (p.18-19, Sec. V.C). We also added two brief explanations in the revised manuscript (p.9, right column, 1st paragraph).

Fig. R5. Effects of GKAP modifications on the mechanical behavior of the 6xPSD condensate. (A) Comparison of the measured force relaxation curves $F(t)/F_0$ between the 6xPSD with pi-GKAP (red circles) and the control (un-phosphorylated 6xPSD, black squares).

(B) Comparison of the measured $F(t)/F_0$ among the 6xPSD with one DLS substitution in GKAP (6xPSD-1xDLS, red circles), 6xPSD with three DLS substitutions in GKAP (6xPSD-3xDLS, blue triangles), and the control (6xPSD with wild-type GKAP, black squares).

Additional questions/comments:

1. Are all 6 proteins required for phase separation in the PSD system? Or are just the scaffold proteins necessary? It would be great if you could introduce the 6xPSD a little bit more in this paper – specifically, which proteins are required – and summarize the main points from prior work that the reader should be aware of.

In the previous study of the 6xPSD system (Zeng, et al., Cell, **174**(5), 1172 (2018)), it has been shown that among the six protein components, PSD-95, GKAP, Shank3, and Homer3 are the core scaffold proteins that drive the phase separation and further recruit NR2B (glutamate receptor) and SynGAP (an enzyme). While two protein components, such as PSD-95 and SynGAP, can induce phase separation at higher protein concentrations, the four scaffold proteins combined together are the main drivers for phase separation at physiologically relevant concentrations.

Following the referee's suggestion, we added a new introduction paragraph on the 6xPSD system in the Introduction (p.3, left column, 2nd paragraph). We added a new figure (Fig. 1(A)) to illustrate what neuronal synapse and postsynaptic density (PSD) are.

2. In the 2nd paragraph in the introduction, when citing works that using particle tracking, at least two important works should be cited in addition to the current ref's 15-19, specifically Zhang et al 2015 DOI: 10.1016/j.molcel.2015.09.017 and Elbaum-Garfinkle et al 2015 10.1073/pnas.1504822112

The two references (Refs. 13 and 25) were added to the main text as suggested (p.1, left column, 2nd paragraph).

3. Is this really a “newly developed technology” ?

Indeed, atomic force microscopy (AFM) has been used widely in previous studies of biomaterials. Most of the studies, however, assumed that the mechanical property of the biomaterials is purely elastic (time-independent) and only an apparent (time-independent) modulus is obtained using the Hertz model. Here, we demonstrate that the protein condensates are viscoelastic (time-dependent) at the mesoscale, and a new methodology is developed to measure the relaxation (or time-dependent) modulus $E(t)$ and understand its molecular origin.

Following the referee's suggestion, we remove the phrase “newly developed technology” and change it to “using atomic force microscopy (AFM) based mesoscale rheology and quantitative fluorescence measurements of protein mobility and composition” (p.2, left column, 2nd paragraph).

4. I don't see a description for how droplets are made/triggered to form? Are all the components simply mixed?

Yes. The phase separation of 6xPSD is induced by simply mixing the six protein components in the buffer solution at the designated concentration of 5 μM each without adding any crowding reagent.

Following the referee's suggestion, the above sentence was added to the revised manuscript (p.3, left column, 2nd paragraph) and SI (p.1, right column, the last paragraph of Sec. I.A).

5. In the discussion of mesh size estimates, should also cite Wei et al 2017 DOI: 10.1038/nchem.2803

This reference (Ref. 65) was added in the main text as suggested (p.7, left column, 1st paragraph).

6. What is the reference for the mesh size calculation used?

The following three references were used in the estimate of the network mesh size:

1) Richbourg, N. R., & Peppas, N. A. (2020). The swollen polymer network hypothesis: Quantitative models of hydrogel swelling, stiffness, and solute transport. *Prog. Polym. Sci.*, **105**, 101243.

- 2) Tsuji, Y., Li, X., & Shibayama, M. (2018). Evaluation of Mesh Size in Model Polymer Networks Consisting of Tetra-Arm and Linear Poly(ethylene glycol)s. *Gels*, **4**(2), 50.
- 3) Wisniewska, M. A., Seland, J. G., & Wang, W. (2018). Determining the scaling of gel mesh size with changing crosslinker concentration using dynamic swelling, rheometry, and PGSE NMR spectroscopy. *J. Appl. Polym. Sci.*, **135**(45), 46695.

With the measured immobile protein concentration n_{im} , one can find the average volume occupied by each network molecule $v_m=1/(N_A n_{im})$, where N_A is the Avogadro's constant. If this volume is assumed to be a sphere with radius R_m , we have $R_m = (3v_m/4\pi)^{1/3} = [3/(4\pi N_A n_{im})]^{1/3}$. The mesh size (or the correlation length) ξ_{net} can be estimated as the average distance between the two adjacent spheres, namely $\xi_{net} \approx 2R_m = 2(3v_m/4\pi)^{1/3} = 2[3/(4\pi N_A n_{im})]^{1/3}$. This simple estimate of network mesh size has been used in the above three references.

Following the referee's suggestion, we added a discussion section and cited the above three references (Refs. 11-13) in SI (p.8, Sec. II.D). A reference to Supplementary Sec. II.D. was added in the main text (p.7, left column, 1st paragraph).

7. Regarding discussion of the exponential relaxation in relation to poroelastic relaxation ... can you elaborate on why it should take longer for solvent to redistribute, than for local mobile protein concentrations around the network? That doesn't seem intuitive to this reviewer!

The difference between poroelastic relaxation and local concentration relaxation, discussed in the main text, lies primarily in the effective distance through which the solvent or solute (protein) relaxes by diffusion. In the poroelastic model, the solvent relaxation under the mechanical compression takes place over the impact region of radius r_{impact} . When an AFM probe of radius R is in contact with a protein droplet at an indentation depth δ (see Fig. 3(D) in the main text for the contact geometry), the value of r_{impact} can be estimated using the Hertz model as: $r_{impact} \approx \sqrt{R\delta}$ (see Refs. 57, 58, 85 in the main text). The relaxation time for the solvent to diffuse through the impact region scales as $\tau_{poro} \approx \frac{r_{impact}^2}{D_p} = R\delta/D_p$, where D_p is the solvent diffusivity. For the setup used in this experiment, we have $R \approx 7.5 \mu\text{m}$, $\delta \approx 0.2 \mu\text{m}$ (see Fig. 1(I) in the main text), $D_p \approx 1 \mu\text{m}^2/\text{s}$ (see Refs. 58,85 in the main text), and the corresponding poroelastic relaxation time $t_{poro} \approx 1.5 \text{ s}$. As discussed in the main text, the diffusive relaxation time associated with local concentration variations of mobile proteins scales as $\tau_1 \approx \xi_{mo}^2/D \approx 2.8 \text{ ms}$, where D is the diffusion coefficient of the protein in the condensate. Because ξ_{mo} is usually much smaller than the size of the impact region, the protein diffusive relaxation time is thus much smaller than the solvent poroelastic relaxation time.

To clarify this point further, we added a discussion section (Sec. III.B., p10-11) and a new Fig. S9 in SI (p.11). We also revised the related discussion in the main text (p.9, right column, last paragraph).

Response to Referee #2:

The authors wish to thank the referee for his/her careful review and constructive suggestions for this work. These suggestions have helped us considerably improve and strengthen the manuscript. The following are our responses to each of the referee's comments (marked in blue). All the changes made in the main text and supplementary materials are marked in red.

Reviewer #2 (Remarks to the Author):

The authors characterize the viscoelastic properties of a protein condensate formed by six different postsynaptic density protein (called 6xPSD), by means of atomic force microscopy and fluorescence microscopy. They first measure the viscoelastic properties of the multi-protein condensate measuring the stress relaxation modulus. To further investigate the underlying protein network, the authors use fluorescence labelling to track the contribution of each component to the viscoelastic properties of the system. Interestingly, they are able to quantify the proportion of mobile vs. immobile proteins within the condensate. The work is concise and well-written, the characterization of the viscoelasticity is quite complete as they provide both general observations and local analysis of each individual component. However, the work may be not applicable to more homogenous liquid-like systems and thus it could deserve publication in Nature Communications after some major changes.

Major Issues:

1- The main problem with this work is the lack of applicability to other systems with less components. The work seems designed to this specific system made of many proteins that form an underlying network. First, I would suggest the authors to justify why they use the system of 6xPSD, some reasons should be provided in the introduction.

As mentioned in the review report, many recent studies of protein condensates have been focused on simple one- or two-component condensates, such as *C. elegans* protein PGL-3 and the human fused in sarcoma (FUS) protein. These condensates have served as model systems for the study of the biochemical and fluid properties of the protein condensates. Because the molecular interactions in these systems are relatively simple, the Maxwell model was found to describe their mechanical behavior well. A unique feature of Maxwell fluids is that when a constant strain is applied, the resulting stress decays exponentially with a single relaxation time. In living cells, however, the cytoplasm contains various proteins, nucleic acids, and lipids. Thus, the condensates inside the cell often involve multiple protein components with complex multivalent interactions. With increasing protein components and interaction complexities, emergent properties, such as forming percolated protein networks, are expected to arise in the protein condensates, leading to novel mechanical behaviors beyond simple Maxwell fluids. A quantitative description of the mechanical properties of functional condensates and a further understanding of their major roles in regulating essential cellular functions are, therefore, urgently needed.

One of the reasons to choose the 6xPSD is that the 6xPSD condensate mimics the actual functional postsynaptic densities (PSDs) in living neurons. As shown in the newly added Fig. 1(A), PSDs are subcellular and membrane-less structures beneath the postsynaptic membrane that play an essential role in neuronal signal transduction. Studying how PSDs self-assemble and respond to mechanical cues is crucial to understanding synapse functions. The 6xPSD condensate is a well-characterized system, and its biochemical properties, including molecular

interaction nodes and their binding affinities (K_d), have been thoroughly documented in a recent study (Ref. 5 of main text). The six protein components in the 6xPSD system have been shown to play essential roles in building the PSD structures and maintaining the integrity and biological functions of neuronal synapses. Therefore, the 6xPSD is an ideal condensate system for the study attempted here to provide a quantitative description of the mechanical properties of a functional multivalent protein condensate, leading to a further understanding of PSD formation and plasticity in living neurons.

Following the referee's suggestion, we revised the introduction section. In particular, we added a general discussion on why the study of multivalent protein condensates is interesting (p.1, right column, last two paragraphs) and a specific discussion on why the study of the PSD condensate in neurons is critical (p.3, left column, 1st and 2nd paragraphs). We added a new figure (Fig. 1(A)) to illustrate what neuronal synapse and postsynaptic density (PSD) are.

Also, the other system they characterize seems rather like the original provided in the main text.

We assume the referee asked about the difference between samples S1 and S2. The two samples are indeed very similar, but they were prepared by two different groups of researchers. Sample S2 was prepared for the previous experiments reported in the 2018 Cell paper (Zeng et al., Cell 174(5), 1172), and Sample S1 was prepared in this experiment following the same experimental protocols. The only difference between the two samples is that the final concentration of NR2B in S2 is 10 μ M instead of 5 μ M in S1.

Following the referee's suggestion, we added an introduction paragraph in SI Sec. VII to explain the similarities and differences between samples S1 and S2 (p.19, right column, 1st paragraph in Sec. VII).

Second, I wonder to what extent this method for characterizing viscoelasticity is applicable to more simple liquid-like protein condensates such as those studied in other works as these:

Jawerth, L., Fischer-Friedrich, E., Saha, S., Wang, J., Franzmann, T., Zhang, X., ... & Jülicher, F. (2020). Protein condensates as aging Maxwell fluids. *Science*, 370(6522), 1317-1323., Fisher, R. S., & Elbaum-Garfinkle, S. (2020).

Boeynaems, S., Holehouse, A. S., Weinhardt, V., Kovacs, D., Van Lindt, J., Larabell, C., ... & Gitler, A. D. (2019). Spontaneous driving forces give rise to protein– RNA condensates with coexisting phases and complex material properties. *Proceedings of the National Academy of Sciences*, 116(16), 7889-7898.

Alshareedah, I., Borchers, W. M., Cohen, S. R., Singh, A., Posey, A. E., Farag, M., ... & Banerjee, P. R. (2023). Sequence-specific interactions determine viscoelasticity and aging dynamics of protein condensates. *bioRxiv*, 2023-04.

Tunable multiphase dynamics of arginine and lysine liquid condensates. *Nature communications*, 11(1), 4628.

The works presented above use different techniques. To what extent the AFM and FRAP can provide compatible results for the systems studied in this work?

To address the referee's concerns, we repeated the AFM measurements for the one-component PGL-3 condensate, which is the same as that used by Jawerth *et al.* (*Science*, 2020) and was prepared similarly. Figure R1 below compares the obtained force relaxation curves between PGL-3 (red circles) and 6xPSD (black squares). It is seen that the measured force relaxation

$F(t)/F_0$ (proportional to the modulus $E(t)$) for the PGL-3 droplets (red dots) decays to zero much faster than that for 6xPSD and can be well described by a simple exponential decay without power-law relaxation. The single exponential decay in the relaxation modulus $E(t)$ is a hallmark of Maxwell fluids (see, e.g., p.168 in Ref. 28 (Doi, *Soft Matter Physics*, 2013), and p.284 in Ref. 29 (Rubinstein and Colby, *Polymer Physics*, 2003)). The results shown in Fig. R1 thus confirm that the PGL-3 condensate is of Maxwell type, as previously reported. They also demonstrate that our AFM methodology of measuring the relaxation modulus $E(t)$ applies to other homogenous liquid-like systems, and the power-law relaxation in the measured $F(t)/F_0$ is an emergent property of 6xPSD.

Fig. R1. Comparison of the normalized force relaxation $F(t)/F_0$ between the PGL-3 droplets (red circles) and 6xPSD droplets (black squares). The solid lines show the fits of Eq. (1) in the main text to the data points. The fit to the PGL-3 data gives $C_1 = 1$, $\tau_1 = 2.8$ ms, and $C_2 = 0$.

Following the referee's suggestion, we added the PGL-3 data in Fig. 1(F) and a discussion paragraph in the revised main text (p.4, left column, 2nd paragraph). Details about the sample preparation were added to the Supplementary Information (SI) (p.1, Sec. I.A, right column, 1st paragraph).

The relaxation modulus $E(t)$ is commonly used to describe the viscoelasticity of polymeric fluids (see Refs. 28-29 mentioned above). Its Fourier transform is directly linked to the complex shear modulus $G^*(\omega)$ (see supplementary Eq. (S15)), which is an alternative way of characterizing viscoelastic materials in the frequency domain. Since the two relaxation modes in Eq. (2) of the main text are well separated in time, using $E(t)$ is more convenient for identifying the molecular origins of the condensate's viscoelasticity. To compare with the previous results mentioned in the review reports, we converted the measured relaxation modulus $E(t)$ to the complex shear modulus $G^*(\omega)$, which was used previously to describe the viscoelastic properties of PGL-3. It is found that the obtained $G^*(\omega)$ for PGL-3 is in good agreement with the previous results by Jawerth *et al.* (*Science*, 2020) and is well described by the Maxwell model in the frequency domain.

In the revised manuscript, we added a discussion on the relation between the relaxation modulus $E(t)$ in the time domain and $G^*(\omega)$ in the frequency domain (p.3, right column, last paragraph). The four references mentioned above were also cited in the revised manuscript (Refs. 24, 28-30). A new Section and a new figure were added in SI (p.8-10, Fig.S8, Sec. III.A).

The system reminds more to a melt of associative polymers rather than a protein condensate due to the underlying protein mesh. You can check the classic works of Rubinstein and Semenov such as Semenov, A. N., & Rubinstein, M. (1998). Thermoreversible gelation in solutions of associative polymers. 1. Statics. *Macromolecules*, 31(4), 1373-1385.

We thank the referee for bringing this interesting work by Semenov and Rubinstein to our attention. While both the 6xPSD condensate and associative polymers can form reversible networks, the protein interactions in the 6xPSD system are far more complicated than those in the associated polymers. The former is a heterogeneous system involving many specific and non-specific protein interactions between folded domains and short-binding motifs. As illustrated in Fig. 3(B), some protein interactions are multivalent and non-pairwise. The associated polymers that Semenov and Rubinstein studied were uniform polymer chains, which only contain a single type of pairwise crosslinkers. With increasing protein components and interaction complexities, we expect such emergent properties as forming percolated protein networks to arise in the protein condensates, leading to novel mechanical behaviors beyond simple associative polymers.

Following the referee's suggestion, we cited the work by Semenov and Rubinstein in the revised manuscript (p.9, right column, last paragraph, Ref. 73).

2- The introduction to the problem is poor, the authors should introduce the problem and the importance of characterizing the viscoelasticity and why to study the system of 6xPSD. The work lack theoretical approaches to the problem or simulation works that have recently contributed to the field. Here there are a few works they can cite:

Devarajan, D. S., Wang, J., Szała-Mendyk, B., Rekhi, S., Nikoubashman, A., Kim, Y. C., & Mittal, J. (2024). Sequence-dependent material properties of biomolecular condensates and their relation to dilute phase conformations. *Nature Communications*, 15.

Tejedor, A. R., Collepardo-Guevara, R., Ramírez, J., & Espinosa, J. R. (2023). Time-dependent material properties of aging biomolecular condensates from different viscoelasticity measurements in molecular dynamics simulations. *The Journal of Physical Chemistry B*, 127(20), 4441-4459.

Tejedor, A. R., Garaizar, A., Ramírez, J., & Espinosa, J. R. (2021). 'RNA modulation of transport properties and stability in phase-separated condensates. *Biophysical Journal*, 120(23), 5169-5186.

Following the referee's suggestion, we revised the introduction section. In particular, we added a general discussion on why the study of multivalent protein condensates is interesting (p.1, right column, last two paragraphs) and a specific discussion on why the study of the PSD condensate in neurons is critical (p.3, left column, 2nd paragraph). We added a new figure (Fig. 1(A)) to illustrate what neuronal synapse and postsynaptic density (PSD) are. In addition, we added a sentence, "Numerical simulations and theoretical approaches were employed to provide physical insights into the molecular interactions in the condensates", in the first paragraph of the introduction section (p.1, 1st paragraph). The three references mentioned above were cited in the revised manuscript (Refs. 16-18).

3- The authors provide a schematic depiction of the protein network structure in Fig. A-B. It seems to be unclear how they infer that specific structure with the results they provide in the current work. Is this something well-known? If so, please provide the reference. Apart from that figure 3 is quite informative despite it does not include any new analysis. In addition, the authors have performed the same analysis to a second sample (S2), bringing these results (not necessarily all of them) to the main text could improve the impact of the work.

The biochemical properties of the 6xPSD system have been thoroughly studied and were reported in the 2018 Cell paper (Ref. 5, Zeng, et al., *Cell*, 174(5), 1172). Details about the pairwise interactions between different protein components can be found in Refs. [5,46,77-80]. To

simplify the presentation without including too many details about the protein-protein interactions, we coarse-grain the multiple domains of a protein chain as a note, and the complex protein-protein interactions are presented as a simple note-to-note interaction with a dissociation constant K_d . This is shown in Fig. 3(A)-3(B). Following the referee's suggestion, we included Refs. [5, 46,77-80] in the Figure 3 caption (p.8).

We agree with the referee's suggestion to move some of the results for Sample S2 to the main text, which would improve the impact of the work. The only concern we have is that after all the revisions, the page number of the main text has increased to over ten pages, which may already reach the page limit. If the editor agrees, we could move Fig S17 (or only (A) and (C)) and associated discussions to the main text. In the meantime, we added an introduction paragraph in SI Sec. VII to let the reader know the similarities and differences between Samples S1 and S2 (p.19, right column, 1st paragraph in Sec. VII).

4- The work does not estimate the viscosity of the condensate. In the introduction the authors include a discussion about the complex modulus and the crossover of G' and G'' in the low frequency regime when the condensate behaves as a gel. Could the authors estimate the elastic and viscous modulus of the studied system? See for instance:

Alshareedah, I., Borchers, W. M., Cohen, S. R., Singh, A., Posey, A. E., Farag, M., ... & Banerjee, P. R. (2023). Sequence-specific interactions determine viscoelasticity and aging dynamics of protein condensates. *bioRxiv*, 2023-04.

Additionally, the authors can cite the book of Rubinstein (Polymer Physics) as a reference for theoretical expressions.

Following the referee's suggestion, we converted the measured relaxation modulus $E(t)$ to the complex shear modulus $G^*(\omega)$ for both PGL-3 and 6xPSD. As mentioned in our answer to Question #1, the Fourier transform of $E(t)$ is directly linked to the complex shear modulus $G^*(\omega)$ via supplementary Eq. (S15). It is seen from the newly added Fig. S8(A) that the obtained $G'(\omega)$ and $G''(\omega)$ for PGL-3 are well described by the Maxwell model. From the fitting to the Maxwell model (Eq. S16), we obtained the plateau modulus $G_0 \approx 455$ Pa and the crossover time $\tau_1 \approx 2.1$ ms. The corresponding value of the condensate viscosity is $\eta \approx G_0\tau_1/2\pi \approx 0.15$ Pa·s (see Jawerth *et al.* *Science*, 2020 and Doi, *Soft Matter Physics*, 2023). These fitting results are in good agreement with the reference values of $G_0 \approx 500$ Pa, $\tau_1 \approx 3$ ms, and $\eta \approx 0.24$ Pa·s, as reported by Jawerth *et al.* (*Science*, 2020) for fresh PGL-3 samples. Figure S13(A) thus verifies that the PGL-3 condensate is of Maxwell type and our AFM methodology of measuring the relaxation modulus $E(t)$ works.

Similarly, we obtained $G'(\omega)$ and $G''(\omega)$ for the 6xPSD condensate, as shown in the newly added Fig. S8(B). The measure $G'(\omega)$ and $G''(\omega)$ for the 6xPSD condensate cannot be fitted to the Maxwell model. Nevertheless, we can still find the plateau modulus $G_0 \approx 3517$ Pa from the measured $G'(\omega)$ and the crossover time $\tau_1 \approx 3.8$ ms between the measured $G'(\omega)$ and $G''(\omega)$. Hence, the viscosity of the 6xPSD condensate is estimated as $\eta \approx G_0\tau_1/2\pi \approx 2.1$ Pa·s. This value of η for the 6xPSD condensate is about 14 times larger than that of the fresh PGL-3 condensate.

Following the referee's suggestion, we added a discussion paragraph on the relation between the relaxation modulus $E(t)$ in the time domain and $G^*(\omega)$ in the frequency domain in the revised manuscript (p.4, left column, first two paragraphs). The two references mentioned above were cited in the revised manuscript (Refs. 30, 34). A new Section (Sec. III.A) and a new

figure (Fig. S8) were added in SI (p.9). The measured values of the condensate viscosity for PGL-3 and 6xPSD as discussed above were reported in SI Sec. III.A.

5- In Fig. 1C the exponential and potential decay are confusing. Is that a fit to the data? The sum of both contributions clearly does not fit the experimental data. Authors should be clearer on that. The discussion of this figure is confusing. Maybe split the figure in two could help to guide the reader. Just out of curiosity, could you calculate the hysteresis from Fig. 1C as a function of the loading speed? This could be a complementary analysis.

We thank the referee for pointing out the confusing illustration in Fig. 1(C). It was intended to indicate that the initial delay of the measured $E(t)$ can be described by a simple exponential decay with a constant baseline and that a power-law relaxation can describe the long-time decay of the measured $E(t)$. We revised Fig. 1 and removed Fig. 1(C) in the revised manuscript to avoid further confusion.

We assume the referee asked about the hysteresis from Fig. 1(E) of the original manuscript (Fig. 1(H) in the revised Fig. 1). Following the referee's suggestion, we calculated the hysteresis H_A , defined as the area enclosed by the approach and retraction curves in the force-indentation loop (grey area in Fig. R2(A) below). It is seen from Fig. R2(B) below that the obtained A_H depends very weakly on the indentation speed v , as the mean value of A_H changes only slightly for different values of v . To further verify that the obtained hysteresis A_H is not affected by the variations of the maximal force F_0 applied and the maximal indentation δ_0 achieved during the measurements (see Fig. 1(H) in the main text), we plot, in Fig. R2(C) below, the normalized hysteresis $A_H/(F_0 \delta_0)$ as a function of indentation speed v . It is seen that the mean value of the obtained $A_H/(F_0 \delta_0)$ only increases slightly with v .

Following the referee's suggestion, we added the above analysis in SI (p.18, right column, Sec. VI.B). Figure R2 below was also included in SI (p.17, Fig. S17).

Fig. R2. Measured hysteresis in the force-indentation loop. (A) Hysteresis H_A defined as the area enclosed by the approach and retraction curves (grey area) in the force-indentation loop. (B) Measured H_A as a function of indentation speed v . (C) Normalized hysteresis $H_A/(F_0 \delta_0)$ as a function of indentation speed v . The individual data points are obtained from different 6xPSD droplets. The solid symbols show the mean value and the standard deviation of the measurements.

Minor comments:

- In some cases, the notation is a bit confusing. Again, I don't understand why they introduce the complex modulus if they don't perform any analysis in that respect. Could they note the relaxation modulus as $G(t)$ instead of $E(t)$?

In the experiment, what we measured is the compressive modulus (or Young's modulus) $E(t)$ rather than the shear modulus $G(t)$. For homogenous materials in the linear regime, such as the protein condensate studied here, $E(t)$ and $G(t)$ are linked by the simple linear relation, $E = 2(1 + \nu)G$, where ν is the Poisson ratio of the material. Given their high water content, the protein condensates are incompressible in volume, so their Poisson ratio is 0.5 (see Ref. 33, Doi, Soft Matter Physics, 2013). Hence, we have $E = 3G$ for the 6xPSD condensate.

Following the referee's suggestion, we converted the measured relaxation modulus $E(t)$ to the complex shear modulus $G^*(\omega)$ for the 6xPSD condensate (see our answers to Questions #1 and #4 above). In the revised manuscript, we added a discussion on the relation between the relaxation modulus $E(t)$ in the time domain and $G^*(\omega)$ in the frequency domain (p.3, right column, last paragraph). A new Section (Sec. III.A) and a new figure (Fig. S8) were added in SI (p.9).

- In Eq. (3) it would help the reader to provide the explicit expression for the correction factor $C(t)$.

We added the explicit expression of $C(t)$ in Eq. (4) in the main text as suggested (p.4).

- The authors state that the fit of Eq. (4) lead to a universal master curve (Fig. S2) when plotting the $I(t)/A$ vs. t/τ . This is trivial as long as the data can be fitted to Eq. 4, so it is more valuable to discuss why the data follow Eq. (4).

The referee is correct in pointing out that the scaling plot shown in Fig. S2 (Fig. S1 in the revised SI) is an alternative way to plot the data following Eq. (4). The main purpose of Fig. S2 is to show that our choice of the cut-off time $t/\tau = 3$ is sufficiently long to determine the saturation value A for the labeled protein. Figure S2 also revealed that all the data sets with different protein components fit Eq. (4) equally well.

Following the referee's suggestion, we revised the discussion of Fig. S2 to clarify the above point. In the main text, we added a sentence explaining why the FRAP data follow Eq. (5) (p.7, left column, 2nd paragraph).

Response to Referee #4:

The authors wish to thank the referee for his/her careful review and critical comments on this work. The following are our responses to each of the referee's comments (marked in blue). All the changes made in the main text and supplementary materials are marked in red.

Reviewer #4 (Remarks to the Author):

This manuscript discusses probes the mechanical response of a six-component biomolecular condensate using AFM and FRAP.

AFM relaxation measurements reveal a short term (<few ms) exponential relaxation followed by a power-law decay over a few decades. Force-indentation measurements show a rate dependent indentation with a strong adhesive force on pull-off. FRAP experiments reveal a significant immobile fraction inside the droplets. The authors conclude that the mechanical response is viscoelastic, with a short-time Maxwell-like response due to the diffusion of the mobile species, and a long-time power-law response reflecting slow network re-arrangements.

While the data appears to be of high quality, I am skeptical of its physical interpretation.

- AFM a) AFM data has contributions from interfacial forces b) the compressional deformation of AFM mixes poroelastic and viscoelastic responses, c) measured forces are very sensitive to the contact geometry, which is unobserved and d) the data is acquired in a limit where the contact radius is comparable or even bigger than the sample thickness, where results are hard to interpret and very sensitive to this ratio. It seems that particle tracking microrheology would be a much more appropriate choice for probing viscoelasticity.

In the following, we provide brief answers to each of the questions mentioned above. More detailed explanations and the additional experiments conducted to address the referee's specific concerns will be provided in the answers to the follow-up questions.

- a) We conducted additional droplet coalescence experiments to measure the interfacial tension γ of the 6xPSD droplets in water. It is found that the capillary force resulting from the interfacial tension γ is negligibly small compared to the measured force relaxation $F(t)$ and indentation $F(\delta)$ shown in Fig. 1 of the main text. The additional measurements of the contact geometry between the AFM probe and the upper surface of the condensate droplet further confirm that the probe-droplet contact during the force relaxation measurements remains unchanged and that the force-indentation measurements during the loading period exhibit a typical Hertzian contact behavior. Therefore, the contributions to the measured $F(t)$ and $F(\delta)$ must result from the bulk of the 6xPSD droplets and not from interfacial forces.
- b) An important assumption that the poroelastic model made is that the porous matrix of the medium remains stable and does not relax during the poroelastic transport. This may be true for certain polymeric networks, such as strong hydrogels and rubber-like materials, in which the lifetime of their cross-links is longer than the poroelastic relaxation time τ_p . The protein network in the 6xPSD condensate, on the other hand, is weak and dynamic compared with covalent bonds and its cross-links bind and unbind continuously over a range of times, which gives rise to a separate stress relaxation channel that allows the network to flow at long times. As shown in Fig. 1(F) of the main text, the measured $F(t)/F_0$ (i.e., the accumulated stress) has decayed more than 95% through the power-law relaxation over a time span of 0.01--1 s, whereas the poroelastic relaxation has a typical relaxation

time in the range of 1--10 s. Therefore, the poroelastic model is not applicable to dynamic protein networks, such as that in the 6xPSD condensate, whose lifetime is shorter than the poroelastic relaxation time τ_p .

- c) Following the referee's suggestion, we conducted additional measurements of the contact geometry between the AFM probe and the upper surface of the condensate droplet. It is found that the probe-droplet contact during the force relaxation measurements remains unchanged and that the force-indentation measurements during the loading period exhibit a typical Hertzian contact behavior. Our AFM measurements indicated that the adhesion does not affect the force-indentation curves $F(\delta)$ very much during the loading (advancing) period and only appears in the retraction process.
- d) The reason we chose the AFM probe size to be comparable with the droplet size (10–20 μm) is to accurately characterize the modulus of protein condensates at the whole-droplet level with a well-defined measurement geometry and precise mathematical modeling of viscoelastic relaxation. This is particularly important for those protein condensates that involve multiple components and multivalent interactions, such as 6xPSD, in which local inhomogeneity could be present. To further test the technique, we performed additional AFM measurements on a well-characterized model one-component protein condensate, PGL-3, which has been studied previously by Jawerth et al. using optical-tweezer-based active micro-rheology (Science, 2020 (Ref. 24)). The PGL-3 droplets have a similar size range, interfacial tension, and adhesion to those for the 6xPSD droplets. Our AFM results in the time domain fully replicate the findings from the active micro-rheology measurements conducted in the frequency domain. This correspondence between the two rheological measurements is a direct and conclusive validation of our AFM methodology for measuring the relaxation modulus $E(t)$ over a five-decade time span ranging from 0.1ms to 10s. Since the two relaxation modes in Eq. (2) of the main text are well separated in time, using the relaxation modulus $E(t)$ is more convenient for identifying the molecular origins of the condensate's viscoelasticity.

- I think the FRAP data is tricky to interpret because a) the sample is very thin and interactions of the protein with the probe or surface of the sample holder could immobilize protein.

First, the FRAP measurements were conducted separately from the AFM measurements, so the AFM probe does not influence the FRAP measurements. Second, as mentioned in the main text (p.3, right column, 1st paragraph), the height of the condensate droplets is typically 2 μm , which is not very thin compared with the confocal imaging section thickness of $\sim 0.3 \mu\text{m}$. In the FRAP measurements, the vertical position of the confocal imaging section (i.e., the z-position of the imaging plane) was set at the middle plane of the condensate droplets, which is away from the substrate, as illustrated in the figure below. This is a standard protocol and common practice for the FRAP experiments to avoid possible substrate effects on the FRAP.

To clarify these points, we added a brief discussion on the vertical optical sectioning of the FRAP in SI (p.2, Sec. I.B, left column, 2nd paragraph). The phrase “we conduct additional (separate) measurements of FRAP” was added in the main text (p.6, left column, 2nd paragraph).

I am skeptical of the hypothesized relevance to cells, as described at the end of the conclusion. The main issue is that the system has six components (plus buffer). Physical properties will vary dramatically across this high-dimensional phase diagram. There is no description in this manuscript of how the composition of the protein droplets in these studies compares to those in cells.

One of the reasons for choosing the 6xPSD in this study is that the 6xPSD condensate mimics the actual functional excitatory postsynaptic densities (PSDs) in living neurons. The molecular origin of the six protein components is from human neuronal synapses. The 6xPSD condensate is a well-characterized system, and its biochemical and biological functions have been fully documented in recent studies (see Refs. 5, 47, 50 in the main text). The six protein components in the 6xPSD system have been shown to play essential roles in building the PSD structures and maintaining the integrity and biological functions of neuronal synapses (see Refs. 5, 47 in the main text). Among the six protein components, PSD-95, GKAP, Shank3, and Homer3 are the most abundant scaffold proteins in human synapses. They are included in the 6xPSD system as the core scaffold proteins to drive the phase separation along with a synaptic receptor (NR2B) and a PSD enzyme (SynGAP). The 6xPSD is, therefore, directly and closely related to the living neurons. It is an ideal condensate system for the study attempted here to provide a quantitative description of the mechanical properties of a functional multivalent protein condensate, leading to a further understanding of PSD formation and plasticity in living neurons. The 6xPSD system also serves as a prototype for many other multi-component protein condensates in living cells.

The referee is correct in pointing out that the actual functional condensates in living cells involve even more protein components and complex protein interactions, and so do the actual PSDs in living neurons. This is why we chose the 6xPSD system instead of the highly idealized one- or two-component protein condensates, which only involve simple protein interactions and behave as Maxwell fluids. The actual PSDs in living neurons have a size in sub-micrometers (~ 500 nm) and are extremely thin (~ 50 nm) (see Mary Kennedy et al., *Science*, **290** (5492), 750 (2000)). The small size and strong ties to the cell membrane and cytoskeleton cortex make it extremely difficult to directly probe the material properties of the PSDs in living neurons. Our *in vitro* experiment involving six functional protein components and multivalent interactions is the first step in this direction.

While our *in vitro* measurements are limited to a six-component condensate, we found that the relaxation modulus $E(t)$ of the 6xPSD droplets exhibits properties similar to those of the living eukaryotic cells (see Ref. 58 in the main text for the measured $E(t)$ for 10 different cell types). The measured $E(t)$ for both systems contains short-time exponential and long-time power-law relaxation. This similarity in the relaxation modes is caused by a similar coarse-grained mechanical structure of the two systems: mobile proteins (those in the cytosol for living cells) partitioned by a percolated (immobile) protein network (cytoskeletal networks for living cells). Because the typical mesh size of the protein network in the 6xPSD is comparable to that of the cytoskeleton network, the exponential relaxation in the two systems has a similar diffusive relaxation time of 3-5 ms. The power-law relaxation in the two systems arises from the slow relaxation of the (deformed) disordered protein network (or cytoskeleton). While increasing protein components and interaction complexities may change some details of the protein network (and hence the five relaxation parameters in Eq. (2) of the main text), we believe that the overall functional form of the relaxation modulus $E(t)$ (i.e., the two relaxation modes) will remain so long the percolated protein network can survive across the high-dimensional

complex energy landscape, as demonstrated by the 10 different cell types studied in Ref. 58 in the main text.

Following the referee's suggestion, we revised the introduction section. In particular, we added a general discussion on why the study of multivalent protein condensates is related to cell biology (p.1, right column, last two paragraphs) and a specific discussion on why the study of the PSD condensate in neurons is critical (p.3, left column, 2nd paragraph). We added a new figure (Fig. 1(A)) to illustrate what the neuronal synapse and postsynaptic density (PSD) are. In addition, we revised the discussion on the relevance to the living cells in Conclusion (p.10, left column, last two paragraphs).

Further concerns on the physical interpretation of the data

1. I am concerned that the AFM method is sensitive not only to bulk, but also surface properties of the droplets.

a. The determination of the time dependent modulus assumes that the contact area is constant. But contact lines are infamously slow to relax. There is no reason, a priori, to assume that the dynamics of relaxation are due to a bulk relaxation, instead of contact line motion. Do the authors have data on the contact area over time?

To address the referee's concerns, we conducted the droplet coalescence experiment to measure the interfacial tension γ of the 6xPSD droplets in water. The coalescence measurements were carried out using the same AFM setup together with bright field imaging. The AFM tip moved around to perturb the fluid and put two 6xPSD droplets in contact. Under the action of interfacial tension γ , the two condensate droplets gradually merge into a single spherical droplet to minimize the surface energy, as shown in Fig. R1(A) below. A good measure of the surface area reduction is the droplet aspect ratio Γ during the coalescence. Figure R1(B) below shows the obtained $\Gamma(t)$ as a function of coalescence time t . The experimental data are well described by the exponential decay function, $\Gamma(t) = 1 + A_0 \exp[-t/\tau_a]$ (red solid line), where $A_0 (=1.1)$ is the initial amplitude and $\tau_a (=96\text{ s})$ is the characteristic coalescence time. The value of τ_a is determined by, $\tau_a \approx R\eta/\gamma$, where η is the viscosity of the droplet and R is its radius. Figure R1(C) below summarizes the obtained values of η/γ from 33 different fusion events of the 6xPSD droplets, from which we find $\eta/\gamma = 12.1 \pm 7.6\text{ s}/\mu\text{m}$ (mean \pm standard deviation).

Fig. R1. **Droplet coalescence measurement.** (A) Time lapse images showing the merging of two 6xPSD condensate droplets into a single spherical droplet. Scale bar: $15\ \mu\text{m}$. (B) Time-dependence of the measured aspect ratio of the two droplets in contact in (A) (black dots). The red solid line shows an exponential fit to the data. (C) The obtained values of $\eta/\gamma = 12.1 \pm 7.6\text{ s}/\mu\text{m}$ (black diamonds) from multiple 6xPSD droplets.

Using the measured effective viscosity $\eta \approx 2.1\text{ Pa}\cdot\text{s}$ for the 6xPSD droplets, as described in SI Sec. III.A, we obtain the interfacial tension of the 6xPSD droplets in water, $\gamma \approx 0.2 \times 10^{-6}\text{ N/m}$. An estimated value of $\gamma \leq 3 \times 10^{-6}\text{ N/m}$ was also reported for the PGL-3 condensate (see Refs.

16 and 17 in SI). For $\gamma \approx 0.2 \times 10^{-6}$ N/m, the largest possible capillary force acting on the AFM probe is $\pi d\gamma \approx 9.4$ pN, where we have used the AFM probe diameter $d = 15$ μm to estimate the contact line length. The actual contact line length is much smaller than πd , because the indentation δ involved is only a fraction of a micrometer (see Fig. 1 in the main text). As shown in Fig. 1 of the main text, the measured force relaxation $F(t)$ and force-indentation $F(\delta)$ are all in the 1-3 nN range. Because the capillary force is much smaller than the obtained $F(t)$ and $F(\delta)$ shown in Fig. 1 of the main text, it becomes evident that contributions to the obtained $F(t)$ and $F(\delta)$ must result from the bulk of the 6xPSD droplets and the contact line effect on the obtained $F(t)$ and $F(\delta)$ is negligibly minor.

Given our contact geometry, as shown in the figure on the right, the contact line relaxation can be written as $F_{CT}(t) = 2\pi a\gamma \cos\theta(t)$, where $2\pi a$ is the contact line length, γ is the interfacial tension of the 6xPSD droplet in water, and $\theta(t)$ is the local contact angle. As discussed above, the capillary force amplitude (i.e., the contact line force acting on the AFM probe) $2\pi a\gamma$ is smaller than $\pi d\gamma \approx 9.4$ pN. The contact angle variation $\cos\theta(t)$ is bounded below unity. As a result, the contact line relaxation is simply too small to account for the measured force relaxation $F(t)$, which is in the 1-3 nN range.

To make these points clearer, we added a new section (Sec. IV) and a new figure (Fig. S10) to the SI (p.11-12) to discuss the droplet coalescence experiment and the determination of the interfacial tension γ of the 6xPSD droplets in water. We also added a description of the surface tension of the 6xPSD in the revised manuscript (p.5, right column, 1st paragraph).

b. The authors observe adhesive forces comparable to the total force on indentation. I appreciate the control experiments with different surface coatings, but I am not convinced by it. Given the huge adhesion, it's clear that the droplet shape and contact area must be strongly affected by interfacial forces in ways that are not accounted for by the Hertz model

To address the referee's concerns about the contact geometry between the AFM probe and the upper surface of the condensate droplet, we designed a new contact visualization apparatus using confocal microscopy together with a micro-manipulation probe (with the AFM setup as shown in Fig. 1(B) of the main text, we are unable to observe the contact geometry directly). The figure on the right shows the experimental setup. The micro-manipulation probe contains a micron-sized glass fiber (probe holder) with its tip forged into a glass sphere (alternatively, one may glue a glass bead to the fiber tip). A three-axis micro-manipulator driven by stepper motors is used to control the motion of the probe. The 6xPSD condensate solution is placed on a coverslip for imaging and the probe can move around in the solution until it touches a desired condensate droplet. To visualize the probe-droplet contact geometry, we use confocal microscopy to capture the time-lapse and z-stack images of the droplet in contact with the probe having a z-interval of 0.5 μm . The 6xPSD condensate droplets are labeled with a fluorescent dye Cy3, as described in SI Sec. I.

Because the probe is not labeled, the probe-droplet contact region appears as a dark dent in the confocal image. This is clearly shown in the 3D confocal reconstruction images (see newly added Fig. S12 in the SI, p.13). To examine the time dependence of the contact region, we took confocal fluorescent images at three different times: before the contact, right after the contact, and 7 seconds after the contact. We extract quantitative parameters associated with the contact geometry from the obtained confocal images shown in newly added Figs. S12 and S13 (p.13-14, SI). As shown in the figure on the right, the geometry parameters at the probe-droplet contact include the probe radius R_p , the droplet radius R_d , the reduced radius $R=1/(1/R_p+1/R_d)$, the distance δ_1 between the upper surface of the droplet and the bottom of the imaging section, the distance δ_2 between the imaging section bottom and the bottom of the probe, the total indentation $\delta = \delta_1 + \delta_2$, the observed contact radius a , and the calculated Hertzian radius $a_H = (R\delta)^{1/2}$.

Droplet	R_p (μm)	R_d (μm)	R (μm)	δ_1 (μm)	δ_2 (μm)	δ (μm)	a (μm)	a_H (μm)
1	10.0 ± 0.5	13 ± 1	5.7 ± 0.3	3.5 ± 0.5	0.3 ± 0.5	3.8 ± 0.7	4.8 ± 1.0	4.7 ± 0.9
2	10.0 ± 0.5	14 ± 1	5.7 ± 0.3	3.0 ± 0.5	1.0 ± 0.5	4.0 ± 0.7	5.3 ± 1.0	4.8 ± 0.9
3	10.0 ± 0.5	13 ± 1	5.6 ± 0.3	3.5 ± 0.5	0.8 ± 0.5	4.3 ± 0.7	5.0 ± 1.0	4.9 ± 0.9

Table S2 (p.13, SI) in the above summarizes the results of the eight geometry parameters obtained from three 6xPSD condensate droplets. As shown in the above figure, the contact between the probe and the condensate droplet is characterized primarily by two quantitative parameters: the total indentation δ in the vertical direction and the contact radius a in the horizontal direction. For the probe radius $R_p = 10 \mu\text{m}$ comparable to the droplet radius R_d , we find from Table S2 that the measured contact radius a agrees with the calculated Hertzian radius a_H within the experimental uncertainties.

This result suggests that the probe-droplet contact is a Hertzian contact during the loading period. To further verify this conclusion, we show a log-log plot of the measured force-indentation curves $F(\delta)$ at different loading speeds in the figure above. It is seen that the measured force-indentation curves $F(\delta)$ at large indentations (away from the contact point) all follow the Hertz scaling law, $F \propto \delta^{3/2}$, for different loading speeds (or different times after the contact).

Droplet	Time	R_p (μm)	R_d (μm)	R (μm)	δ_1 (μm)	δ_2 (μm)	δ (μm)	a (μm)	a_H (μm)
1	Before contact	10.0 ± 0.5	13 ± 1	5.7 ± 0.3	0	0	0	0	0
	Right after contact	10.0 ± 0.5	13 ± 1	5.7 ± 0.3	3.5 ± 0.5	0.2 ± 0.5	3.7 ± 0.7	4.7 ± 1.0	4.6 ± 0.9
	7s after contact	10.0 ± 0.5	13 ± 1	5.7 ± 0.3	3.5 ± 0.5	0.3 ± 0.5	3.8 ± 0.7	4.8 ± 1.0	4.7 ± 0.9
2	Before contact	10.0 ± 0.5	14 ± 1	5.7 ± 0.3	0	0	0	0	0
	Right after contact	10.0 ± 0.5	14 ± 1	5.7 ± 0.3	3.0 ± 0.5	0.8 ± 0.5	3.8 ± 0.7	5.5 ± 1.0	4.7 ± 0.9
	7s after contact	10.0 ± 0.5	14 ± 1	5.7 ± 0.3	3.0 ± 0.5	1.0 ± 0.5	4.0 ± 0.7	5.3 ± 1.0	4.8 ± 0.9
3	Before contact	10.0 ± 0.5	13 ± 1	5.6 ± 0.3	0	0	0	0	0
	Right after contact	10.0 ± 0.5	13 ± 1	5.6 ± 0.3	3.5 ± 0.5	0.5 ± 0.5	4.0 ± 0.7	5.3 ± 1.0	4.7 ± 0.9
	7s after contact	10.0 ± 0.5	13 ± 1	5.6 ± 0.3	3.5 ± 0.5	0.8 ± 0.5	4.3 ± 0.7	5.0 ± 1.0	4.9 ± 0.9

Table S3 (p.15, SI) above summarizes the results of the eight geometry parameters obtained at three different times: before the contact, right after the contact, and 7 seconds after the contact. It is seen from Table S3 that the contact radius a and the total indentation δ obtained 7 seconds after the contact remain essentially the same as those obtained right after the contact (within the experimental uncertainties) and no visible time dependence is observed for the contact parameters. From the results shown in Tables S2, S3, and the figure above, we conclude that the probe-droplet contact during the force relaxation measurements remains unchanged and that the force-indentation measurements during the loading period exhibit a typical Hertzian contact behavior.

Our AFM measurements indicate that the adhesion does not affect the force-indentation curves $F(\delta)$ very much during the loading (advancing) period and only appears in the retraction process. As mentioned above, the measured force-indentation curves $F(\delta)$ during the loading period can all be well described by the Hertz scaling, $F(\delta) \propto \delta^{3/2}$. This asymmetric effect was also observed in polyacrylamide hydrogels (see Lai *et al.*, *Extreme Mechanics Letters* **31**, 100540 (2019) and *Mechanics of Materials* **159**, 103877 (2021)). The figure on the right (adapted from Fig. 1 in Lai *et al.*, *Extreme Mechanics Letters* **31**, 100540 (2019)) shows an example of the measured force-indentation curve $F(h)$ for a polyacrylamide hydrogel. Again, the measured $F(h)$ during the loading period follows the Hertz scaling law, $F(h) \propto h^{3/2}$ (red dashed line). It is seen from the figure that adhesion only plays a role during the retraction process. While we have not yet found a microscopic theory to explain this asymmetric effect, the experimental observations are nevertheless robust, and we see no reason to dispute the experimental fact.

[Figure Redacted]

If the adhesive force were significant during the loading period, the resulting force-indentation curve $F(\delta)$ would be very different from what we have observed. The figure on the right shows an example of the measured force-indentation curve

$F(\delta)$ on a sticky and soft polydimethylsiloxane (PDMS) substrate (adapted from Fig. 1 in Pham *et al.* *Physical Review Materials*, **1**, 015602 (2017)). Unlike Fig. 1(H) in the main text and the first figure shown above (adapted from Fig. 1 in Lai *et al.*, *Extreme Mechanics Letters* **31**, 100540 (2019)), here adhesion plays a significant role in both the loading (approach) and unloading (retraction) processes.

[Figure Redacted]

Because the probe-droplet contact remains unchanged with time, the measured stress relaxation $E(t)$ will not be affected by the surface adhesion. This conclusion is further confirmed by the additional measurements reported in the supplementary Sec. VI.A. Specifically, we showed in Fig. S15 that the measured force relaxation $F(t)/F_0$ does not change with different surface treatments, the amplitude of the applied force F_0 , and the probe holding time t_{dwell} . As shown in Fig. S16, these three parameters have introduced significant changes in adhesion during the retraction process. Nonetheless, the measured force-indentation curves $F(\delta)$ during the loading period do not change with them. Our experiments thus demonstrate that the measured force

relaxation $F(t)/F_0$ and force-indentation $F(\delta)$ during the loading period are not influenced by the surface adhesion at the probe-droplet contact, which only plays a role in the retraction process. We find no experimental evidence that the Hertzian contact has changed during the loading period, nor has the contact area changed during the force relaxation measurements.

To clarify these points, we added a new section in SI (Sec. V, p.12-15) to discuss the new measurements on contact geometry. This section includes four new figures (Fig. S11-S14) and two new tables (Tables S2 and S3). In addition, we added two summary paragraphs at the end of the discussion about the surface adhesion in SI Sec. VI.A (p.18, left column, last two paragraphs) and cited all the references mentioned above.

2. Poroelasticity is discounted much too quickly. Poroelastic time constants are sensitive to geometry, network properties and solvent properties. So, randomly picking some timescales from poroelasticity papers is insufficient to rule out this mechanism. The best way to rule it out would be to look at relaxation timescale versus indentation depth or radius of indenter.

An important assumption that the poroelastic model made is that the porous matrix of the medium remains stable and does not relax during the poroelastic transport. In other words, the lifetime of the protein network itself needs to be longer than the poroelastic relaxation time τ_p . This may be true for certain polymeric networks, such as strong hydrogels and rubber-like materials, in which the lifetime of their cross-links is longer than τ_p . The protein network in the 6xPSD condensate, on the other hand, is weak and dynamic, and its cross-links bind and unbind continuously over a range of times, which gives rise to a separate stress relaxation channel that allows the network to flow at long times. As shown in Fig. 1(F), the measured $F(t)/F_0$ (i.e., the accumulated stress) has decayed more than 95% through the power-law relaxation over a time span of 0.01--1 s, whereas the poroelastic relaxation has a typical relaxation time in the range of 1--10 s (see the revised SI Sec. III.B for more details about the numerical estimates). Therefore, the poroelastic model is not applicable to transient protein networks, such as that in the 6xPSD condensate, whose lifetime is shorter than the poroelastic relaxation time τ_p .

Following the referee's suggestion, we perform additional force relaxation measurements with varying applied force amplitude F_0 . The increase in F_0 gives rise to an increase in δ and hence an increase in the contact area $a^2 \approx R\delta$, where R is the AFM probe radius. Figure R2 below shows how the measured exponential relaxation time τ_1 changes with δ . It is seen that the obtained values of τ_1 from different 6xPSD droplets (different colors) do not show any systematic variations when the imposed indentation δ is changed nearly five times. The measured values of τ_1 scatter over the range of 4 ± 1 s (the error bar gives a measure of droplet-to-droplet variations) and do not show a clear linear dependence on δ , as expected for poroelastic relaxation. Figure R9 thus confirms that the exponential relaxation in the measured $F(t)/F_0$ for the 6xPSD droplets is not associated with poroelastic relaxation.

Fig. R2. Measured exponential relaxation time τ_1 as a function of indentation δ . The relaxation time τ_1 is obtained from the measured force relaxation curves $F(t)/F_0$ for different values of the applied force amplitude F_0 . Color-coded data points are obtained from different 6xPSD droplets.

To clarify these points, we reorganized the Discussion section and included a new paragraph on poroelastic relaxation at the end of the section (p.9, right column, last paragraph). We also added a new subsection in SI (p.10, Sec. III.B) to discuss the relaxation channels for the 6xPSD condensate. This subsection included Fig. R2 in the above (p.11, Fig. S9).

3. The authors jump too quickly to viscoelasticity. The null hypothesis of liquid-liquid phase separation (which they invoke in the first sentence of the abstract) is that the fluids are Newtonian. The manuscript would be much more compelling if their analysis started by showing that the observed behavior is inconsistent with Newtonian rheology. Given the strength of the interfacial forces (as seen in the adhesion), and likely contact line pinning and relaxation, I think this is a hard case to make. The droplet shape should have an exponential relaxation determined by the competition of its size and the capillary velocity. Contact lines can relax very slowly. Further, because the indenter is so close to the surface, and the sample is so thin, one would have to also rule out slow drainage of the lubricating film. In the absence of measurements of the contact radius versus indentation, it will be hard to rule out Newtonian fluids for the force-indentation curve.

Many previous studies have shown that the protein condensates are not simple Newtonian fluids (e.g., Refs. 5, 17-19, 21, 29-31, 33-34 in the main text). Because protein condensates are concentrated polymeric solutions, they are expected to be non-Newtonian fluids. Recent studies of the material properties of the protein condensates revealed that simple one- or two-component condensates behave as Maxwell fluids (Refs. 21, 22, 24 in the main text). A central finding of our work is that for a functional protein condensate involving multi-components and multivalent interactions, an emergent property arises in the condensate, i.e., forming a percolated protein network, leading to novel mechanical behaviors beyond simple Maxwell fluids. Therefore, the null hypothesis for this study is to conduct similar AFM measurements for a simple one-component condensate, which behaves as a Maxwell fluid without a network. The term “liquid-liquid phase separation” used for studying protein condensates has broadened to include non-Newtonian fluids, such as Maxwell fluids. While understanding the subtle difference between phase separation in a Newtonian fluid and that in a non-Newtonian fluid is interesting in its own right, it is nevertheless beyond the scope of the present study.

With this understanding, we repeated the AFM measurements for the model one-component protein condensate, PGL-3, which behaves as a Maxwell fluid as demonstrated by Jawerth *et al.* using optical–tweezer–based active micro-rheology (Science, 2020 (Ref. 24) and additional Refs. 51, 52 in the main text). The PGL-3 sample was prepared similarly as described in Ref. 24. Figure R3 below compares the obtained force relaxation curves between PGL-3 (red circles) and 6xPSD (black squares). It is seen that the measured force relaxation $F(t)/F_0$ (proportional to the modulus $E(t)$) for the PGL-3 droplets (red dots) decays to zero much faster than that for 6xPSD and can be well described by a simple exponential decay without power-law relaxation. The single exponential decay in the relaxation modulus $E(t)$ is a hallmark of Maxwell fluids (see, p.168 in Ref. 33 and p.284 in Ref. 34). The results shown in Fig. R3 thus confirm that the PGL-3 condensate is of Maxwell type, as reported previously. They also demonstrate that our AFM methodology of measuring the relaxation modulus $E(t)$ works, and the power-law relaxation in the measured $F(t)/F_0$ is an emergent property of the 6xPSD.

Fig. R3. Comparison of the normalized force relaxation $F(t)/F_0$ between the PGL-3 droplets (red circles) and 6xPSD droplets (black squares). The solid lines show the fits of Eq. (1) in the main text to the data points. The fit to the PGL-3 data gives $C_1 = 1$, $\tau_1 = 2.8$ ms, and $C_2 = 0$.

The relaxation modulus $E(t)$ is commonly used to describe the viscoelasticity of polymeric fluids (see Refs. 33, 34 mentioned above). Its Fourier transform is directly linked to the complex shear modulus $G^*(\omega)$ (see supplementary Eq. (S15)), which is an alternative way of characterizing viscoelastic materials in the frequency domain. In the 2020 Science paper by Jawerth *et al.*, the authors used optical-tweezer-based active micro-rheology to measure $G^*(\omega)$. To have a more quantitative comparison with the active micro-rheology measurements, we converted the measured relaxation modulus $E(t)$ to $G^*(\omega)$ using Eq. (S15). It is seen from the newly added Fig. S8(A) (p.9, SI) that the obtained $G'(\omega)$ and $G''(\omega)$ for PGL-3 are well described by the Maxwell model. From the fitting to the Maxwell model (Eq. S16), we obtained the plateau modulus $G_0 \approx 455$ Pa and the crossover time $\tau_1 \approx 2.1$ ms. The corresponding value of the condensate viscosity is $\eta \approx G_0(\tau_1/2\pi) \approx 0.15$ Pa·s. These fitting results agree with the reference values of $G_0 \approx 500$ Pa, $\tau_1 \approx 3$ ms, and $\eta \approx 0.24$ Pa·s., as reported by Jawerth *et al.* (Science, 2020) for fresh PGL-3 samples.

Thus, our AFM results in the time domain fully replicate the findings from the active micro-rheology measurements conducted in the frequency domain. This correspondence between the two rheological measurements is a direct and conclusive validation of our AFM methodology for measuring the relaxation modulus $E(t)$. The various concerns and doubts raised by the referee in the review report, including the effect of interfacial forces, adhesion, contact line pinning and relaxation, variations of the droplet shape and contact area, and slow drainage of a lubricating film, could be equally applicable to the PGL-3 droplets, which have a similar size range, interfacial tension, and adhesion (see Fig. R4 below) to those for the 6xPSD droplets. Yet, our AFM results for the PGL-3 condensate have demonstrated that these hypothetical effects do not affect our AFM results in any visible way, as explained above. Our interpretation

Fig. R4. Measured force-indentation curve $F(\delta)$ for a PGL-3 droplet when the AFM probe approaches (loading curve, red solid line) and retracts (unloading curve, red dotted line) from the droplet surface. The measurement was made at a fixed loading/unloading speed $v = 20$ $\mu\text{m/s}$. The black dash line shows a fit of the Hertz scaling law, $F \propto \delta^{3/2}$, to the loading curve.

of the AFM data in the time domain is entirely consistent with the conventional interpretation of the complex shear modulus $G^*(\omega)$ in the frequency domain obtained using optical-tweezer-based active micro-rheology.

Incidentally, as mentioned in the main text (p.3, right column, first paragraph), the 6xPSD droplets have a typical height $h \approx 2 \mu\text{m}$ and the indentation δ is in the range of $0.3 \mu\text{m}$. Therefore, the gap Δ between the AFM probe and substrate is larger than $1.5 \mu\text{m}$, which is not as small as assumed in the review report. For example, the lubrication force in the buffer solution at room temperature (viscosity $\eta \approx 0.15 \text{ mPa}\cdot\text{s}$) for a gap $\Delta = 1.5 \mu\text{m}$ and probe radius $R = 7.5 \mu\text{m}$ is given by $F_L \approx 6\pi\eta Rv(R/4\Delta)$ (see, Cottin-Bizonne *et al.*, Phys. Rev. Lett. **94**, 056102 (2005)), where v is the loading speed. For the largest loading speed $v = 100 \mu\text{m/s}$ used in the experiment, we have $F_L = 2.5 \text{ pN}$, which is negligibly small compared to the force range of 1-3 nN involved in the force relaxation and indentation measurements.

To make these points clearer, we made several changes in the revised manuscript. First, we added the PGL-3 data in Fig. 1(F) and a discussion paragraph in the main text (p.4, left column, 2nd paragraph). Details about the sample preparation were added to the SI (p.1, Sec. I.A, right column, 1st paragraph). Second, we added a discussion on the relation between the relaxation modulus $E(t)$ and $G^*(\omega)$ in the main text (p.3, right column, last paragraph). A new section (Sec. III.A) and a new figure (Fig. S8) were added to the SI (p.8-10).

At the end of the day, while the experiments are nicely done. I think there are too many ambiguities in the interpretation to give a definitive result. If this was the only method to try to measure these things, I might be more lenient. Since other methods exist to more reliably and easily measure rheology (particle tracking microrheology), I am very reluctant to put much emphasis, as a community, on these results. Do the authors have some microrheology data on this system to compare to? Is there any information in the AFM test that is not available from microrheology?

Up to now, particle tracking microrheology (or optical-tweezer-based active microrheology) has only been used for simple one- or two-component protein condensates, which behave as Maxwell fluids (see Refs. 24, 51, 52 in the main text). For the first time, our work demonstrated that an emergent property arises with increasing protein components and interaction complexities, i.e., forming a percolated protein network in the protein condensate, leading to novel mechanical behaviors beyond simple Maxwell fluids. Our system comprises six essential postsynaptic proteins (6xPSD) from human synapses. The 6xPSD condensate mimics the actual functional excitatory postsynaptic densities in living neurons.

As mentioned in the Answer to Question #3, the active (or passive) micro-rheology measures the complex shear modulus $G^*(\omega)$ in the frequency domain. In contrast, our AFM-based mesoscale rheology measures the relaxation modulus $E(t)$ in the time domain. The two techniques complement each other, as the Fourier transform of $E(t)$ is directly linked to $G^*(\omega)$. To further demonstrate the equivalence of the two techniques, we performed the AFM measurements on the PGL-3 condensate, which is a model one-component protein condensate and has been studied previously by Jawerth *et al.* using optical-tweezer-based active micro-rheology (Science, 2020 (Ref. 24)). Our AFM results in the time domain fully replicate the findings from the active micro-rheology measurements conducted in the frequency domain. This correspondence between the two rheological measurements is a direct and conclusive validation of our AFM methodology for measuring the relaxation modulus $E(t)$.

The various concerns and doubts raised in the review report could be equally applicable to the PGL-3 droplets, which have similar size, interfacial tension, and adhesion to those for the 6xPSD droplets. Yet, our AFM results for the PGL-3 condensate have demonstrated that these hypothetical effects do not affect our AFM results in any visible way, as explained in the above answers to the referee's questions. With the additional experimental evidence, we aim to show that there are no ambiguities in our interpretation of the AFM data in the time domain, which aligns perfectly with the conventional interpretation of the complex shear modulus $G^*(\omega)$ in the frequency domain, as obtained through active or passive micro-rheology. Since the two relaxation modes in Eq. (2) of the main text are well separated in time, using the relaxation modulus $E(t)$ is more convenient for identifying the molecular origins of the condensate's viscoelasticity.